# Recurrent Natural Policy Gradient for POMDPs

**Semih Cayci** *cayci@mathc.rwth-aachen.de*
*Department of Mathematics*
*RWTH Aachen University*

**Atilla Eryilmaz** *eryilmaz.2@osu.edu*
*Department of Electrical and Computer Engineering*
*The Ohio State University*

**Reviewed on OpenReview:** *https://openreview.net/forum?id=6GO1eOvgIf*

## Abstract

Solving partially observable Markov decision processes (POMDPs) remains a fundamental challenge in reinforcement learning (RL), primarily due to the curse of dimensionality induced by the non-stationarity of optimal policies. In this work, we study a natural actor-critic (NAC) algorithm that integrates recurrent neural network (RNN) architectures into a natural policy gradient (NPG) method and a temporal difference (TD) learning method. This framework leverages the representational capacity of RNNs to address non-stationarity in RL to solve POMDPs while retaining the statistical and computational efficiency of natural gradient methods in RL. We provide non-asymptotic theoretical guarantees for this method, including bounds on sample and iteration complexity to achieve global optimality up to function approximation. Additionally, we characterize pathological cases that stem from long-term dependencies, thereby explaining limitations of RNN-based policy optimization for POMDPs.

## 1 Introduction

Reinforcement learning (RL) for partially observable Markov decision processes (POMDPs) has been a particularly challenging problem due to the absence of an optimal stationary policy, which leads to a curse of dimensionality as the space of non-stationary policies grows exponentially over time (Krishnamurthy, 2016; Murphy, 2000). To address this curse of dimensionality in solving POMDPs, finite-memory (Yu & Bertsekas, 2008; Yu, 2012; Kara & Yüksel, 2023; Cayci et al., 2024a) and RNN-based (Lin & Mitchell, 1993; Whitehead & Lin, 1995; Wierstra et al., 2010; Mnih et al., 2014; Ni et al., 2021; Lu et al., 2024) model-free RL approaches are widely used to solve POMDPs. Despite the empirical success of RNN-based model-free RL methods, a rigorous theoretical understanding of their performance in the POMDP setting remains limited.

We begin by outlining two key observations that motivate our approach:

**Observation 1.** Recurrent neural networks (RNNs) have been extensively employed in model-free reinforcement learning (RL) to solve partially observable Markov decision processes (POMDPs) (Whitehead & Lin, 1995; Wierstra et al., 2010; Mnih et al., 2014). Recent work Ni et al. (2021) demonstrates that RNN-based model-free RL can perform competitively with more sophisticated and structured approaches under appropriate hyperparameter and architecture choices. In Lu et al. (2024), shortcomings of emerging transformers in solving POMDPs were demonstrated, and it was shown, somewhat surprisingly, that particular recurrent architectures can achieve superior practical performance in certain scenarios. However, despite this plethora of works that demonstrate the effectiveness of RNN-based model-free algorithms for solving POMDPs, a concrete theoretical understanding of these methods is still in a nascent stage. This is particularly important since, as noted by Ni et al. (2021), RNN-based model-free RL algorithms are sensitive to optimization parameters, and identification of provably good choices is important for practice.

**Observation 2.** Natural policy gradient (NPG) framework has been shown to be effective in solving MDPs due to its versatility in encompassing powerful function approximators, such as deep neural networks (Wang et al., 2019; Cayci et al., 2024b). However, a naïve application of such non-recurrent model-free RL algorithms to solve POMDPs has been observed to be ineffective (Ni et al., 2021), which necessitate careful incorporation of recurrent architectures into the policy optimization framework. This calls for the need to incorporate and analyze policy optimization, particularly NPG framework, augmented with recurrent architectures, to obtain a provably effective solution for POMDPs.

Our study is motivated by these observations and guided by the following key questions, each addressed in this work:

**$Q_1$. How can we achieve (i) provably effective and (ii) computation/memory-efficient policy evaluation for non-stationary policies in partially observable environments?**
▷ A temporal difference (TD) learning algorithm with an IndRNN (Rec-TD) overcomes the so-called *perceptual aliasing* problem imperative in memoryless TD learning for POMDPs (Singh et al., 1994), and achieves *near-optimal* policy evaluation, provided a sufficiently large network (Theorem 5.4 and Remark 5.5). Our analysis identifies the *exploding semi-gradients* pathology in policy evaluation, which can significantly increase network and iteration complexities to mitigate perceptual aliasing under long-term dependencies (Remark 5.6), and demonstrates the role of regularization to mitigate this. We also provide empirical results in random-POMDP instances in Appendix C.

**$Q_2$. How can we parameterize non-stationary policies by a rich and practically feasible class of RNNs and perform efficient policy optimization?**

▷ We represent non-stationary policies using IndRNNs with SOFTMAX parameterization as a form of finite-state controller, and perform computationally efficient NPG updates (based on path-based compatible function approximation for POMDPs) for policy optimization. The policy optimization update (called Rec-NPG) is aided by Rec-TD as the critic (Section 4).

**$Q_3$. What are the memory, computation and sample complexities of the resulting Rec-NAC method, which employs Rec-NPG for policy updates and Rec-TD for policy evaluation?**

▷ Our non-asymptotic analyses of Rec-TD (Theorem 5.4) and Rec-NPG (Theorem 6.3) demonstrate their near-optimality in the large-network limit while highlighting dependencies on memory, long-term POMDP dynamics, and RNN smoothness. Pathological cases with long-term dependencies may require exponentially growing resources (Remarks 5.6-6.4).

These results establish principled and scalable RL solutions for POMDPs, offering insights into the interplay between memory, smoothness, and optimization complexity.

## 1.1 Previous work

Natural policy gradient method, proposed by Kakade (2001), has been extensively investigated for MDPs (Agarwal et al., 2020; Cen et al., 2020; Khodadadian et al., 2021; Liu et al., 2020; Cayci et al., 2024c), and analyses of NPG with feedforward neural networks (FNNs) have been established by Wang et al. (2019); Liu et al. (2019); Cayci et al. (2024b). As these works consider MDPs, the policies are stationary. In our case, the analysis of RNNs and POMDPs constitute a very significant challenge.

Standard TD learning, which does not have a memory structure, was shown to be suboptimal for POMDPs (Singh et al., 1994). We incorporate RNNs into TD learning as a form of memory to address this problem in this work.

In Yu (2012); Singh et al. (1994); Uehara et al. (2022); Kara & Yüksel (2023); Cayci et al. (2024a), finite-memory policies based on sliding-window approximations of the history were investigated. Bilinear frameworks with memory-based policies (Uehara et al., 2022) and Hilbert space embeddings with deterministic

latent dynamics (Uehara et al., 2023) enable sample-efficient learning under specific model structures. In Guo et al. (2022), an offline RL algorithm for the specific class of linear POMDPs was proposed. Unlike these existing works, our approach integrates RNNs with NAC methods, providing a scalable and theoretically grounded framework for general POMDPs without requiring structural assumptions such as deterministic transitions, fixed memory windows, or linear POMDP dynamics. Value- and policy-based model-free RL algorithms based on RNNs have been widely considered in practice to solve POMDPs (Lin & Mitchell, 1993; Whitehead & Lin, 1995; Wierstra et al., 2010; Mnih et al., 2014; Ni et al., 2021; Lu et al., 2024). However, these works are predominantly experimental, thus there is no theoretical analysis of RNN-based RL methods for POMDPs to the best of our knowledge. In this work, we also present theoretical guarantees for RNN-based NPG for POMDPs. For structural results on the hardness of RL for POMDPs, we refer to (Liu et al., 2022; Singh et al., 1994).

## 1.2 Notation

For a finite set $\mathbb{A}$, $\Delta(\mathbb{A}) = \{v \in \mathbb{R}_{\geq 0}^{|\mathbb{A}|} : \sum_{a \in \mathbb{A}} v_a = 1\}$ is the set of probability vectors over the set $\mathbb{A}$. $\mathsf{Rad}(\alpha) = \mathsf{Unif}\{-\alpha, \alpha\}$ for $\alpha \in \mathbb{R}_+$.

## 2 Preliminaries on Partially Observable Markov Decision Processes

In this paper, we consider a discrete-time infinite-horizon partially observable Markov decision process (POMDP) with the (nonlinear) dynamics

$$\mathbb{P}(S_{t+1} = s | S_k, A_k, k \leq t) =: \mathcal{P}((S_t, A_t), s),$$
$$\mathbb{P}(Y_t = y | S_t) =: \phi(S_t, y),$$

for any $s \in \mathbb{S}$ and $y \in \mathbb{Y}$, where $S_t$ is an $\mathbb{S}$-valued *state*, $Y_t$ is a $\mathbb{Y}$-valued *observation*, and $A_t$ is an $\mathbb{A}$-valued *control* process with the stochastic kernels $\mathcal{P} : \mathbb{S} \times \mathbb{A} \times \mathbb{S} \to [0, 1]$ and $\phi : \mathbb{S} \times \mathbb{Y} \to [0, 1]$. We consider finite but arbitrarily large $\mathbb{A}, \mathbb{Y}$ and $\mathbb{S}$, where

$$\mathbb{A} \subset \mathbb{R}^{d_1}, \mathbb{Y} \subset \mathbb{R}^{d_2}$$

for some $d_1, d_2 \in \mathbb{Z}_+$ with $d := d_1 + d_2$, and $\|(y, a)\|_2 \leq 1$ for any $(y, a) \in \mathbb{Y} \times \mathbb{A}$. In this setting, the state process $(S_t)_{t \in \mathbb{N}}$ is not observable by the controller. Let

$$Z_t = \begin{cases} Y_0, & \text{if } t = 0, \\ (Z_{t-1}, A_{t-1}, Y_t), & \text{if } t > 0, \end{cases} \tag{1}$$

be the history process, which is available to the controller at time $t \in \mathbb{N}$, and

$$\bar{Z}_t := (Z_t, A_t) = (Y_0, A_0, \ldots, Y_t, A_t), \tag{2}$$

be the history-action process.

**Definition 2.1** (Admissible policy). An admissible control policy $\pi = (\pi_t)_{t \in \mathbb{N}}$ is a sequence of measurable mappings $\pi_t : (\mathbb{Y} \times \mathbb{A})^t \times \mathbb{Y} \to \Delta(\mathbb{A})$, and the control at time $t$ is chosen under $\pi_t$ randomly as

$$\mathbb{P}(A_t = a | Z_t = z_t) = \pi_t(a | z_t),$$

for any $z_t \in (\mathbb{Y} \times \mathbb{A})^t \times \mathbb{Y}$. We denote the class of all admissible policies by $\Pi_{\mathsf{NM}}$.

If an action $a$ is taken at state $s$, then a deterministic reward $r(s, a)$ with $|r(s, a)| \leq r_\infty < \infty$ is obtained.

**Definition 2.2** (Value function, $\mathcal{Q}$-function, advantage function). Let $\pi$ be an admissible policy, and $\mu \in \Delta(\mathbb{Y})$. The value function under $\pi$ with discount factor $\gamma \in (0, 1)$ is defined as

$$\mathcal{V}_t^\pi(z_t) := \mathbb{E}^\pi\Big[\sum_{k=t}^\infty \gamma^{k-t} r(S_k, A_k) \Big| Z_t = z_t\Big], \tag{3}$$

for any $z_t \in (\mathbb{Y} \times \mathbb{A})^t \times \mathbb{Y}$. Similarly, the state-action value function (also known as $\mathcal{Q}$-function) and the advantage function under $\pi$ are defined as

$$
\begin{aligned}
\mathcal{Q}_t^\pi(\bar{z}_t) &:= \mathbb{E}^\pi \Big[ \sum_{k=t}^\infty \gamma^{k-t} r(S_k, A_k) \Big| \bar{Z}_t = \bar{z}_t \Big], \\
\mathcal{A}_t^\pi(z_t, a) &:= \mathcal{Q}_t^\pi(z_t, a) - \mathcal{V}_t^\pi(z_t),
\end{aligned}
\tag{4}
$$

for any $\bar{z}_t \in (\mathbb{Y} \times \mathbb{A})^{t+1}$, respectively.

Given an initial observation distribution $\mu \in \Delta(\mathbb{Y})$, the optimization problem is

$$
\max_{\pi \in \Pi_{\mathsf{NM}}} \mathcal{V}^\pi(\mu),
\tag{5}
$$

where

$$
\mathcal{V}^\pi(\mu) := \sum_{y \in \mathbb{Y}} \mathcal{V}_0^\pi(y_0) \mu(y_0).
$$

We denote an optimal policy as $\pi^\star \in \arg\max\limits_{\pi \in \Pi_{\mathsf{NM}}} \mathcal{V}^\pi(\mu)$.

*Remark* 2.3 (Curse of history in RL for POMDPs). Note that the problem in equation 5 is significantly more challenging than its subcase of (fully-observable) MDPs since there may not exist an optimal stationary policy (Krishnamurthy, 2016; Singh et al., 1994). As such, the policy search is over *non-stationary* randomized policies of type $\pi = (\pi_0, \pi_1, \dots)$ where $\pi_t : (\mathbb{Y} \times \mathbb{A})^t \times \mathbb{Y} \to \Delta(\mathbb{A})$ depends on the history of observations $Z_t = (Y_0, A_0, Y_1, \dots, A_{t-1}, Y_t)$ for $t \in \mathbb{N}$. In this case, direct extensions of the existing reinforcement learning methods for MDPs become intractable, even for finite $\mathbb{Y}, \mathbb{A}$: the memory complexity of a non-stationary policy $\pi \in \Pi_{\mathsf{NM}}$ at epoch $t \in \mathbb{N}$ is $\mathcal{O}(|\mathbb{Y} \times \mathbb{A}|^{t+1})$, growing exponentially.

In the following section, we formally introduce the RNN architecture that we study in this paper.

## 3 Independently Recurrent Neural Network Architecture

We consider an independently recurrent neural network (IndRNN) architecture in this work (Li et al., 2018; 2019). This architecture has been featured in POPGym (Morad et al., 2023) as it enables RNNs with large sequence lengths by handling long dependencies in practical applications. In other works, it has been shown to be effective for POMDPs in practice as well (Lu et al., 2024; Elelimy et al., 2024).

Let $X_t = (Y_t, A_t) \in \mathbb{R}^d$, therefore $\bar{Z}_t = (X_0, X_1, \dots, X_t)$ for any $t \in \mathbb{Z}_+$ by equation 2. The central structure in an IndRNN is the sequence of hidden states $H_t = (H_t^{(1)}, H_2^{(2)}, \dots, H_t^{(m)}) \in \mathbb{R}^m$ for $t = 0, 1, \dots$, which evolves according to

$$
H_t^{(i)}(\bar{Z}_t; \mathbf{W}, \mathbf{U}) = \varrho\Big( W_{ii} H_{t-1}^{(i)}(\bar{Z}_{t-1}; \mathbf{W}, \mathbf{U}) + \langle U_i, X_t \rangle \Big) \text{ for all } i \in [m],
\tag{6}
$$

with the initial condition $H_0^{(i)}(\bar{Z}_0; \mathbf{W}, \mathbf{U}) := \varrho(\langle U_i, X_0 \rangle)$, where $\varrho : \mathbb{R} \to \mathbb{R}$ is a smooth activation function, $\mathbf{W} = \mathrm{diag}(W_{11}, W_{22}, \dots, W_{mm})$ and $\mathbf{U}$ is an $m \times d$ matrix whose $i$-th row is $U_i^\top$ for $i \in [m]$. We assume a smooth activation function $\varrho$ with $|\varrho(z)| \leq \varrho_0, |\varrho'(z)| \leq \varrho_1$ and $|\varrho''(z)| \leq \varrho_2$ for all $z \in \mathbb{R}$, which is satisfied by many widely-used activation functions including tanh and the sigmoid function. We consider a linear readout layer with weights $c \in \mathbb{R}^m$, which leads to the output

$$
F_t(\bar{Z}_t; \mathbf{W}, \mathbf{U}, c) = \frac{1}{\sqrt{m}} \sum_{i=1}^m c_i H_t^{(i)}(\bar{Z}_t; \mathbf{W}, \mathbf{U}).
\tag{7}
$$

The operation of an independently recurrent neural network is illustrated in Figure 1. Following the neural tangent kernel literature, we omit the task of training the linear output layer $c \in \mathbb{R}^m$ for simplicity, and study the training dynamics of $(\mathbf{W}, \mathbf{U})$, which is the main challenge (Du et al., 2018; Oymak & Soltanolkotabi,

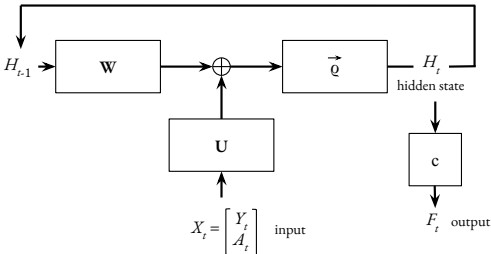

Figure 1: An independently recurrent neural network (IndRNN) in the RL context.

2020; Cai et al., 2019; Wang et al., 2019). Consequently, we denote the learnable parameters of an IndRNN compactly in the vector form as

$$
\Theta = \begin{pmatrix} \Theta_1 \\ \Theta_2 \\ \vdots \\ \Theta_m \end{pmatrix} \in \mathbb{R}^{m(d+1)} \text{ where } \Theta_i = \begin{pmatrix} W_{ii} \\ U_i \end{pmatrix} \in \mathbb{R}^{d+1} \text{ for } i \in [m]. \tag{8}
$$

We use $\Theta$ and $(\mathbf{W}, \mathbf{U})$ interchangeably throughout the paper.

A key feature of the neural tangent kernel analysis is the random initialization (Bai & Lee, 2019; Chizat et al., 2019; Cayci et al., 2023).

**Definition 3.1** (Symmetric random initialization). Let $(c^0, \Theta^0) = (c_i^0, \Theta_i^0)_{i \in [m]}$ be a random vector such that

$$
c_i^0 \overset{\text{iid}}{\sim} \mathsf{Rad}(1),
$$
$$
\Theta_i^0 := \begin{pmatrix} W_{ii}^0 \\ U_i^0 \end{pmatrix} \overset{\text{iid}}{\sim} \begin{pmatrix} \mathsf{Rad}(\alpha) \\ \mathcal{N}(0, I_d) \end{pmatrix},
$$
$$
c_{i+m/2}^0 = -c_i^0 \text{ and } \Theta_{i+m/2}^0 = \Theta_i^0
$$

for $i = 1, 2, \ldots, \frac{m}{2}$. We call $(c^0, \Theta^0)$ a symmetric random initialization, and denote the distribution of $(c^0, \Theta^0)$ as $\zeta_0$.

For both policy optimization (Algorithm 1) and policy evaluation (Algorithm 2), the IndRNNs are randomly initialized according to Definition 3.1. Such random initialization schemes are widely adopted in practice, and play a fundamental role in the theoretical analysis of deep learning algorithms Bai & Lee (2019); Chizat et al. (2019); Wang et al. (2019); Cai et al. (2019); Liu et al. (2019).

In the following subsection, we define the reference function class determined by overparameterized IndRNNs in a detailed way, which will be instrumental in the theoretical results and their analyses. We note that this subsection can be skipped for those who would like to focus on the algorithmic design.

## 3.1 Reference Function Class for Independently Recurrent Neural Networks

A fundamental question in reinforcement learning with function approximation is to determine a concrete reference function class for the function approximation architecture that is used for approximation in the value and policy spaces (Bertsekas & Tsitsiklis, 1996). In this subsection, we will identify and discuss the reference function class defined by the IndRNN architecture that will be used for incorporating memory to solve POMDPs. In order to motivate the discussion, we first overview basic reference function classes for (fully-observable) MDPs, then extend the discussion to POMDPs.

**Function approximation in MDPs.** Let us consider value-based reinforcement learning in the case of MDPs, where the objective is to learn the Q-function under a given stationary policy $\pi$. The approximation

error for a given reference class $\mathscr{F}$ of functions $f : \mathbb{S} \times \mathbb{A} \to \mathbb{R}$ is

$$\epsilon_{\text{app}}(\mathscr{F}) := \inf_{f \in \mathscr{F}} \mathbb{E}_{s,a}[(\mathcal{Q}^{\pi}(s,a) - f(s,a))^2]. \tag{9}$$

For example, if a linear function approximation scheme with a given feature map $\phi : \mathbb{S} \times \mathbb{A} \to \mathbb{R}^p$ is used, then the reference function class is $\mathscr{F} := \{(s,a) \mapsto \theta^\top \phi(s,a) : \theta \in \mathbb{R}^p\} = \text{span}(\Phi)$ where $\Phi := [\phi^\top(s,a)]_{s,a}$ is the feature matrix. In the case of linear MDPs Jin et al. (2020), we have $\mathcal{Q}^{\pi} \in \mathscr{F}$ and $\epsilon_{\text{app}}(\mathscr{F}) = 0$; otherwise TD(0) with this linear approximation scheme has an inevitable approximation error $\frac{1}{1-\gamma} \epsilon_{\text{app}}(\mathscr{F})$ (Bertsekas & Tsitsiklis, 1996). The reference function class for a randomly-initialized single hidden-layer feedforward neural network with frozen output layer is

$$\mathscr{F}_{\text{NTK}} := \{(s,a) \mapsto \mathbb{E}_{u_0 \sim \mathcal{N}(0,I_d)}[\boldsymbol{v}(u_0)^\top \nabla_u \varrho(\langle(s,a), u_0\rangle)] \text{ such that } \mathbb{E}_{u_0 \sim \mathcal{N}(0,I_d)}[\|\boldsymbol{v}(u_0)\|_2^2] < \infty\}, \tag{10}$$

where $\boldsymbol{v} : \mathbb{R}^d \to \mathbb{R}^d$ (Liu et al., 2019; Wang et al., 2019; Cayci et al., 2023). Technically, the completion of $\mathscr{F}_{\text{NTK}}$ yields the reproducing kernel Hilbert space (RKHS) of the so-called neural tangent kernel

$$\kappa(x, x') := \mathbb{E}_{u_0}[\nabla_u^\top \varrho(u_0^\top x) \nabla_u \varrho(u_0^\top x')] = x^\top x' \mathbb{E}[\varrho'(u_0^\top x)\varrho'(u_0^\top x')] \text{ for any } x, x' \in \mathbb{S} \times \mathbb{A}$$

and its explicit analysis shows that it is provably rich (Ji et al., 2019). For a detailed discussion on the function space $\mathscr{F}_{\text{NTK}}$ and its role in reinforcement learning, we refer to Section A.2 in Liu et al. (2019) and Cayci et al. (2024b). Due to the concrete approximation bounds for $\mathscr{F}_{\text{NTK}}$, the representational assumption $\mathcal{Q}^{\pi} \in \mathscr{F}_{\text{NTK}}$ is standard in the theoretical analyses of neural TD learning for MDPs, and the objective is to prove that neural TD learning can learn any $\mathcal{Q}^{\pi} \in \mathscr{F}_{\text{NTK}}$ using samples with finite-time and finite-sample guarantees (Cai et al., 2019; Wang et al., 2019; Cai et al., 2019; Cayci et al., 2023). Without the representational assumption $\mathcal{Q}^{\pi} \in \mathscr{F}_{\text{NTK}}$, the optimality guarantees in Cai et al. (2019); Liu et al. (2019); Wang et al. (2019); Cayci et al. (2023) hold up to an additional error term proportional to $\frac{1}{1-\gamma} \epsilon_{\text{app}}(\mathscr{F}_{\text{NTK}})$.

**Function approximation in RL for POMDPs.** Analogous to the approximation error analysis in RL for MDPs discussed earlier, our objective here is to identify a suitable reference function class for the IndRNN architecture defined in equation 7. Building on the framework of Cayci & Eryilmaz (2025), we present an infinite-width characterization of IndRNNs in the neural tangent kernel (NTK) regime. This directly extends the reference function class $\mathscr{F}_{\text{NTK}}$ in 10 for feedforward neural networks in the neural RL literature (Cai et al., 2019; Wang et al., 2019; Cayci et al., 2024b) to the partially observable setting with recurrent models. We note that our reference function class reduces to the feedforward neural networks as a specific case (see Remark 3.4).

For any $t \in \mathbb{Z}_+$ and input $\bar{Z}$, symmetric initialization ensures that $F_t(\bar{Z}; \Theta^0) = 0$. Furthermore, the first-order Taylor expansion of $F_t$ at $\Theta \in \mathbb{R}^{m(d+1)}$ around $\Theta^0$ yields

$$F_t(\bar{Z}; \Theta) = \nabla_\Theta^\top F_t(\bar{Z}; \Theta^0)(\Theta - \Theta^0) + \mathcal{O}\left(\frac{\|\Theta - \Theta^0\|^2}{\sqrt{m}}\right). \tag{11}$$

As $m \to \infty$, the linear part $\nabla_\Theta^\top F_t(\bar{Z}; \Theta^0)(\Theta - \Theta^0)$ is able to approximate a rich class of functions determined by the reproducing kernel Hilbert space (RKHS) of the recurrent neural tangent kernel defined as

$$\kappa_t(\bar{Z}, \bar{Z}') := \lim_{m \to \infty} \nabla_\Theta^\top F_t(\bar{Z}; \Theta^0) \nabla_\Theta F_t(\bar{Z}; \Theta^0),$$

for $t \in \mathbb{Z}_+$. In the following, we characterize this sequence of reproducing kernel Hilbert spaces for $t \in \mathbb{Z}_+$ explicitly, following Cayci & Eryilmaz (2025).

Let $w_0 \sim \text{Rad}(\alpha)$ and $u_0 \sim \mathcal{N}(0, I_d)$ be independent random variables, and $\theta_0 := (w_0, u_0)$. Given a sequence $\bar{\boldsymbol{z}} = (x_0, x_1, \dots) \in (\mathbb{Y} \times \mathbb{A})^{\mathbb{Z}_+}$, let

$$h_t(\bar{z}_t; \theta_0) := \varrho(w_0 h_{t-1}(\bar{z}_{t-1}; \theta_0) + \langle u_0, x_t \rangle) \text{ for } t = 0, 1, 2, \dots,$$

with the initial condition $h_{-1} := 0$. and

$$\mathcal{I}_t(\bar{z}_t; \theta_0) := \varrho'(w_0 h_{t-1}(\bar{z}_{t-1}; \theta_0) + \langle u_0, x_t \rangle).$$

Then, the neural tangent random feature mapping[1] at time $t$ is defined as

$$\psi_t(\bar{z}_t; \theta_0) := \sum_{k=0}^{t} w_0^k \begin{pmatrix} h_{t-k-1}(\bar{z}_{t-k-1}; \theta_0) \\ x_{t-k} \end{pmatrix} \prod_{j=0}^{k} \mathcal{I}_{t-j}(\bar{z}_{t-j}; \theta_0),$$

Based on the sequence of neural tangent random features, the neural tangent random feature matrix is defined as $\Psi(\bar{z}; \theta_0) = \Psi_\infty(\bar{z}; \theta_0)$, where

$$\Psi_T(\bar{z}; \theta_0) := \begin{pmatrix} \psi_0^\top(\bar{z}_0; \theta_0) \\ \psi_1^\top(\bar{z}_1; \theta_0) \\ \vdots \\ \psi_{T-1}^\top(\bar{z}_{T-1}; \theta_0) \end{pmatrix}, \tag{12}$$

for any $T \in \mathbb{Z}_+$.

**Definition 3.2** (Transportation mapping). Let $\mathscr{H}$ be the set of mappings $\boldsymbol{v} : \mathbb{R}^{1+d} \to \mathbb{R}^{1+d}$ such that $\boldsymbol{v}(\theta_0) := \begin{pmatrix} v_w(\theta_0) \\ v_u(\theta_0) \end{pmatrix}$ for $\theta_0 = (w_0, u_0)$ with $\mathbb{E}[\|\boldsymbol{v}(\theta_0)\|_2^2] < \infty$, where $w_0 \sim \mathsf{Rad}(\alpha)$ and $u_0 \sim \mathcal{N}(0, I_d)$. We call $\boldsymbol{v} \in \mathscr{H}$ a transportation mapping, following Ji & Telgarsky (2019); Ji et al. (2019).

**Definition 3.3** (Reference function class for IndRNNs). We define the reference function class of IndRNNs for any sequence-length $T \geq 1$ as

$$\mathscr{F}_T := \left\{ \bar{z} \mapsto \mathbb{E}\left[\Psi_T(\bar{z}; \theta_0)\boldsymbol{v}(\theta_0)\right] = \begin{pmatrix} f_0^\star(\bar{z}_0; \boldsymbol{v}) \\ \vdots \\ f_{T-1}^\star(\bar{z}_{T-1}; \boldsymbol{v}) \end{pmatrix} : \boldsymbol{v} \in \mathscr{H}, \bar{z} \in (\mathbb{Y} \times \mathbb{A})^{\mathbb{Z}_+} \right\},$$

where $f_t^\star(\bar{z}_t; \boldsymbol{v}) := \mathbb{E}[\psi_t^\top(\bar{z}_t; \theta_0)\boldsymbol{v}(\theta_0)]$ for any $\bar{z} \in (\mathbb{Y} \times \mathbb{A})^{\mathbb{Z}_+}$. The same transportation mapping $\boldsymbol{v}$ is used to define $f_t^\star$ for all $t \in \mathbb{N}$, which is a characteristic feature of weight-sharing in RNNs. We denote $\mathscr{F} := \mathscr{F}_\infty$.

*Remark* 3.4 (Reduction to $\mathscr{F}_{\mathrm{NTK}}$). Note that setting $T = 1$ yields the random feature map

$$\psi_t(\bar{z}_0; \theta_0) = \begin{pmatrix} 0 \\ \nabla_u \varrho(\langle u_0, x_0 \rangle) \end{pmatrix},$$

since $\nabla_u \varrho(\langle x_0, u_0 \rangle) = x_0 \varrho'(\langle x_0, u_0 \rangle)$. Hence, for any $\boldsymbol{v} \in \mathscr{H}$, we have

$$\mathscr{F}_1 = \{x_0 \mapsto \mathbb{E}[\boldsymbol{v}_u(u_0)^\top \nabla_u \varrho(\langle x_0, u_0 \rangle)] : \mathbb{E}\|\boldsymbol{v}_u(u_0)\|_2^2 < \infty\},$$

which is exactly the reference function class $\mathscr{F}_{\mathrm{NTK}}$ for feedforward neural networks given in equation 10. In other words, $\{\mathscr{F}_T : T \in \mathbb{Z}_+\}$ contains $\mathscr{F}_{\mathrm{NTK}}$ with $\mathscr{F}_1 = \mathscr{F}_{\mathrm{NTK}}$, which is the reference function class in neural RL literature for MDPs (Wang et al., 2019; Liu et al., 2019). $\mathscr{F}_1$ is dense in the space of continuous functions on a compact set (Ji et al., 2019).

*Remark* 3.5 (Fully-connected RNNs). IndRNNs utilize a diagonal hidden-to-hidden weight matrix $\mathbf{W}$, which was shown to be very effective in handling long-term dependencies in RL compared to conventional RNNs, GRU and LSTM architectures (Morad et al., 2023). In addition to its practical benefits, IndRNNs have theoretical niceties as well, as they enable (i) explicit characterization of the reference function class, and (ii) direct control and analysis of the spectral radius of W. Both of these theoretical amenities are lost when $\mathbf{W}$ does not inherit a diagonal structure.

## 3.2 Max-Norm Projection for IndRNNs

Given an initialization $(\mathbf{W}(0), \mathbf{U}(0), c)$ as in Definition 3.1 and a vector $\rho = (\rho_\mathsf{w}, \rho_\mathsf{u})^\top \in \mathbb{R}_{>0}^2$ of projection radii, we define the compactly-supported set of weights $\Omega_{\rho,m} \subset \mathbb{R}^{m(d+1)}$ as

$$\Omega_{\rho,m} = \left\{\Theta \in \mathbb{R}^{m(d+1)} : \max_i |W_{ii} - W_{ii}(0)| \leq \frac{\rho_\mathsf{w}}{\sqrt{m}}, \ \max_i \|U_i - U_i(0)\| \leq \frac{\rho_\mathsf{u}}{\sqrt{m}}\right\}. \tag{13}$$

---

[1] The feature uses a complicated weighted-sum of all past inputs $x_k, k \leq t$, leading to a discounted memory to tackle non-stationarity. $x_{t-k}$ is scaled with $w_0^k \sim \mathsf{Rad}(\alpha)$, thus it yields a fading memory approximation of the history if $\alpha < 1$.

Given any symmetric random initialization $(\mathbf{W}(0), \mathbf{U}(0), c)$ and $\rho \in \mathbb{R}^2_{>0}$, the set $\Omega_{\rho,m}$ is a compact and convex subset of $\mathbb{R}^{m(d+1)}$, and for any $\Theta \in \Omega_{\rho,m}$, we have

$$\max_{1 \leq i \leq m} |W_{ii} - W_{ii}(0)| \leq \frac{\rho_{\mathsf{w}}}{\sqrt{m}},$$

$$\max_{1 \leq i \leq m} \|U_i - U_i(0)\| \leq \frac{\rho_{\mathsf{u}}}{\sqrt{m}}.$$

Let

$$\mathbf{Proj}_{\Omega_{\rho,m}}[\Theta] = \left[ \underset{w \in \mathcal{B}_2\left(W_{ii}(0), \frac{\rho_{\mathsf{w}}}{\sqrt{m}}\right)}{\arg\min} |W_{ii} - w_i|, \quad \underset{u_i \in \mathcal{B}_2\left(U_i(0), \frac{\rho_{\mathsf{u}}}{\sqrt{m}}\right)}{\arg\min} \|\mathbf{U}_i - u_i\|_2 \right]_{i \in [m]} \tag{14}$$

As such, the projection operator $\mathbf{Proj}_{\Omega_{\rho,m}}[\cdot]$ onto $\Omega_{\rho,m}$ is called the max-norm projection (or regularization) (Goodfellow et al., 2013; Srebro et al., 2004). As an immediate consequence, $\Theta \in \Omega_{\rho,m}$ implies that $|W_{ii}| \leq |W_{ii} - W_{ii}(0)| + |W_{ii}(0)| \leq \alpha + \frac{\rho_{\mathsf{w}}}{\sqrt{m}} =: \alpha_m$, which implies a strict control over $\max_{i \in [m]} |W_{ii}|$. As we will see in Section 5 and Section 6, such a strict control over the norm of the hidden-to-hidden weights $W_{ii}$ has a significant importance in stabilizing the training of IndRNNs. Similar projection mechanisms for IndRNNs are adopted in practice as well (Morad et al., 2023). For further details, we refer to Appendix A.

## 4 Rec-NAC: A High-Level Algorithmic View

In this section, we present a high-level description of our Recurrent Natural Actor-Critic (Rec-NAC) Algorithm with two inner loops, critic (called Rec-TD) and actor (called Rec-NPG), for policy optimization with RNNs. The details of the inner loops of the algorithm will be given in the succeeding sections. We use an admissible policy $\pi = (\pi_t)_{t \in \mathbb{N}}$ that is parameterized by a recurrent neural network $(F_t(\cdot; \Phi))_{t \in \mathbb{N}}$ of the form given in equation 7 with a network width $m \in \mathbb{Z}_+$. To that end, for any $t \in \mathbb{N}$, let

$$\pi_t^\Phi(a|z_t) := \frac{\exp\left(F_t((z_t, a); \Phi)\right)}{\sum_{a' \in \mathbb{A}} \exp\left(F_t((z_t, a'); \Phi)\right)}, \tag{15}$$

for any $z_t \in (\mathbb{Y} \times \mathbb{A})^t \times \mathbb{Y}$ and $a \in \mathbb{A}$ with the parameter $\Phi \in \mathbb{R}^{m(d+1)}$. The high-level operation of Rec-NAC is summarized in Algorithm 1.

---

**Algorithm 1** Recurrent Natural Actor-Critic (Rec-NAC) – a High-level description

1: Initialize the actor RNN as $(c, \Phi(0)) \sim \zeta_0$ (see Definition 3.1).
2: **for** $n = 0, 1, 2, \ldots, N - 1$ **do**
3:     **Critic.** Independently initialize the weights of the critic IndRNN as $(c^n, \Theta^n(0)) \overset{\text{iid}}{\sim} \zeta_0$.
4:         Run Rec-TD in Algorithm 2 for $K_{\mathsf{td}}$ iterations, and obtain $\bar{\Theta}^n := K_{\mathsf{td}}^{-1} \sum_{k < K_{\mathsf{td}}} \Theta^n(k)$
5:         Estimate $\mathcal{Q}_t^{\pi^{\Phi(n)}}$ by $\hat{\mathcal{Q}}_t^{(n)}(\cdot) := F_t(\cdot; \bar{\Theta}^n)$ for all $t < T$.
6:     **Actor.** Apply projected-SGD to obtain

$$\omega_n \in \underset{\omega \in \Omega_{\rho,m}}{\arg\min} \; \mathbb{E}_\mu^\pi \left[ \sum_{t=0}^{T-1} \gamma^t \left( \nabla \ln \pi_t^n(A_t|Z_t)\omega - \hat{\mathcal{A}}_t^{(n)}(\bar{Z}_t) \right)^2 \right],$$

7:         where the estimated advantage function is

$$\hat{\mathcal{A}}_t^{(n)}(z_t, a) := \hat{\mathcal{Q}}_t^{(n)}(z_t, a) - \hat{\mathcal{V}}_t^{(n)}(\bar{Z}_t),$$

8:         for $\hat{\mathcal{Q}}_t^{(n)}(\cdot) := F_t(\cdot; \bar{\Theta}^n)$ and $\hat{\mathcal{V}}_t^{(n)}(\cdot) := \sum_{a' \in \mathbb{A}} \pi_t^{\Phi(n)}(a'|z_t)\hat{\mathcal{Q}}_t^{(n)}(\cdot, a')$.
9:     **Policy update.**

$$\Phi(n+1) = \Phi(n) + \eta \cdot \omega_n.$$

10: **end for**

---

For information regarding the algorithmic tools, i.e., random initialization and max-norm regularization for RNNs, we refer to Section A.

In the following two sections, we derive the critic (Section 5) and the actor (Section 6) in full detail, and provide concrete performance bounds for these methods in each section.

# 5 Critic: Recurrent Temporal Difference Learning ($\mathrm{Rec}$-TD)

In this section, we study a policy evaluation method for POMDPs, which will serve as the critic.

**Policy evaluation problem.** Consider the policy evaluation problem for POMDPs under a given admissible policy $\pi \in \Pi_{\mathsf{NM}}$. Given an initial observation distribution $\mu \in \Delta(\mathbb{Y})$, policy evaluation aims to solve

$$\min_{\Theta \in \Omega_{\rho,m}} \mathcal{R}_T^\pi(\Theta) := \mathbb{E}_\mu^\pi \left[ \sum_{t=0}^{T-1} \gamma^t \Big( F_t(\bar{Z}_t; \Theta) - \mathcal{Q}_t^\pi(\bar{Z}_t) \Big)^2 \right], \tag{16}$$

where $T \in \mathbb{N}$ is the sequence length (i.e., the length of the truncated trajectory $\bar{Z}$), and $\{F_t : t \in \mathbb{N}\}$ is an IndRNN given in equation 7 – we drop the superscript $\mathsf{a}$ for simplicity throughout the discussion. The expectation in $\mathcal{R}_T^\pi(\Theta)$ is with respect to the joint probability law $P_T^{\pi,\mu}$ of the stochastic process $\{(S_t, A_t, Y_t) : t \in [0,T]\}$ where $Z_0 \sim \mu$.

## 5.1 Recurrent TD Learning Algorithm

In this section, we present a multi-step temporal difference learning algorithm for computing the sequence of state-action value functions $\{\mathcal{Q}_t^\pi : t \in \mathbb{N}\}$ for large POMDPs.

We assume access to a sampling oracle capable of generating independent trajectories from a given initial state distribution (Bhandari et al., 2018; Cai et al., 2019).

**Assumption 5.1** (Sampling oracle). Given an initial state distribution $\mu$, we assume that the system can be independently started from $S_0 \sim \mu$, i.e., independent trajectories $\{(S_t, Y_t, A_t) : t \in [T]\} \sim P_T^{\pi,\mu}$ are obtained.

Rec-TD is presented in Algorithm 2. We study the performance of Rec-TD numerically in Section C under long-term and short-term dependencies to validate our theoretical results in Section 5.2.

*Remark* 5.2 (Intuition behind Rec-TD). In a stochastic optimization setting, the loss-minimization for $\mathcal{R}_T(\Theta)$ would be solved by using gradient descent, where the gradient is

$$\nabla_\Theta \mathcal{R}_T^\pi(\Theta) = 2\mathbb{E}_\mu^\pi \left[ \sum_{t=0}^{T-1} \gamma^t \Big( F_t(\bar{Z}_t; \Theta) - \mathcal{Q}_t^\pi(\bar{Z}_t) \Big) \nabla F_t(\bar{Z}_t; \Theta) \right].$$

On the other hand, the target function $\mathcal{Q}_t^\pi$ is unknown and to be learned. Following the bootstrapping idea for MDPs in Sutton (1988), we exploit an extended *non-stationary Bellman equation* in Proposition B.3, and use $r_t + \gamma F_{t+1}(\bar{Z}_{t+1}; \Theta)$ as a bootstrap estimate for the unknown $\mathcal{Q}_t^\pi(\bar{Z}_t)$. Note that, in the realizable case with $F_t(\cdot; \Theta^\star) = \mathcal{Q}_t^\pi(\cdot)$, $t \in \mathbb{Z}_+$ for some $\Theta^\star$, we have $\mathbb{E}_\mu^\pi[\check{\nabla}\mathcal{R}_T(\bar{Z}_T; \Theta^\star)] = 0$, motivating the use of the stochastic approximation in this partially observable setting.

## 5.2 Theoretical Analysis of Rec-TD: Finite-Time Bounds and Global Near-Optimality

In the following, we prove that Rec-TD with max-norm regularization achieves global optimality in expectation. To characterize the impact of long-term dependencies on the performance of Rec-TD, let $p_t(x) = \sum_{k=0}^{t-1} |x|^k$, and $q_t(x) = \sum_{k=0}^{t-1} (k+1)|x|^k$, $x \in \mathbb{R}, t \in \mathbb{N}$.

In the following, we present a regularity condition on the state-action value functions.

**Assumption 5.3** (Regularity of $(\mathcal{Q}_t^\pi)_t$). $\{\mathcal{Q}_t^\pi : t \in \mathbb{N}\} \in \mathscr{F}$ with a transportation mapping $\boldsymbol{v} = (v_w, v_u) \in \mathscr{H}$ such that $\sup_{\theta \in \mathbb{R}^{d+1}} \|v_u(\theta)\|_2 \le \nu_{\mathsf{u}}$ and $\sup_{\theta \in \mathbb{R}^{d+1}} |v_w(\theta)| \le \nu_{\mathsf{w}}$.

---

**Algorithm 2** Recurrent TD Learning Algorithm

---

1: **Input:** step-size $\eta > 0$, max-norm projection radius $\rho = (\rho_w, \rho_u)$, sequence-length $T$.
2: Initialize $(c, \Theta(0)) \sim \zeta_0$ according to Definition 3.1.
3: **for** $k = 0, 1, 2, \ldots, K - 1$ **do**
4:     Sample an initial state $S_0^k \sim \mu$ independently.
5:     Observe $Y_0^k \sim \Phi(S_0^k, \cdot)$.
6:     Choose an action $A_0^k \sim \pi_0(\cdot | Z_0^k)$.
7:     Set $\check{\nabla}\mathcal{R}_T^k := 0$.
8:     **for** $t = 0, 1, \ldots, T$ **do**
9:         State transition $S_{t+1}^k \sim \mathcal{P}((S_t^k, A_t^k), \cdot)$.
10:        Observe $Y_{t+1}^k \sim \Phi(S_{t+1}^k, \cdot)$.
11:        Choose an action $A_{t+1}^k \sim \pi_{t+1}(\cdot | Z_{t+1}^k)$.
12:        Compute temporal difference $\delta_t(\bar{Z}_t^k, \Theta(k))$ where

$$\delta_t(\bar{z}_{t+1}; \Theta) := r_t + \gamma F_{t+1}(\bar{z}_{t+1}; \Theta) - F_t(\bar{z}_t; \Theta).$$

13:        Update stochastic semi-gradient:

$$\check{\nabla}\mathcal{R}_T^k \leftarrow \check{\nabla}\mathcal{R}_T^k + \gamma^t \delta_t(\bar{Z}_{t+1}^k; \Theta(k)).$$

14:     **end for**
15:     Parameter update with max-norm projection

$$\Theta(k + 1) = \mathbf{Proj}_{\Omega_{\rho,m}}\left[\Theta(k) + \eta \cdot \check{\nabla}\mathcal{R}_T^k\right].$$

16: **end for**

---

Assumption 5.3 is a representational assumption, stating that $(\mathcal{Q}_t^\pi)_t$ lies in the RKHS induced by the random features $\Psi_T(\bar{z}; \theta_0)$ defined in equation 12. It directly extends Assumption 4.1 in Wang et al. (2019) and Assumption 2 in Cayci et al. (2024b) to POMDPs, and exactly recovers these assumptions when $T = 1$ (see Remark 3.4).

**Theorem 5.4** (Finite-time bounds for Rec-TD). *Under Assumptions 5.1-5.3, for any projection radius $\rho \succeq \nu = (\nu_{\mathsf{w}}, \nu_{\mathsf{u}})$ and step-size $\eta > 0$, Rec-TD with max-norm projection achieves the following error bound:*

$$\mathbb{E}\left[\frac{1}{K}\sum_{k=0}^{K-1}\mathcal{R}_T^\pi(\Theta(k))\right] \leq \frac{1}{\sqrt{K}}\left(\frac{\|\nu\|_2^2}{(1-\gamma)} + \frac{C_T^{(1)}}{(1-\gamma)^3}\right) + \frac{C_T^{(2)}}{(1-\gamma)^2\sqrt{m}} + \underbrace{\frac{\gamma^T}{(1-\gamma)K}\sum_{k=0}^{K-1}\omega_{T,k}^2}_{(\heartsuit)}. \quad (17)$$

*for any $K \in \mathbb{N}$, where*

$$C_T^{(1)}, C_T^{(2)} = \mathsf{poly}\left(p_T((\alpha + \rho_w m^{-1/2})\varrho_1), \|\rho\|_2, \|\nu\|_2\right),$$

*are instance-dependent constants that do not depend on $K$, and $\omega_{t,k} := \sqrt{\mathbb{E}[(F_t(\bar{Z}_t; \Theta(k)) - \mathcal{Q}_t^\pi(\bar{Z}_t^k))^2]}$ is a uniformly bounded sequence for $t, k \in \mathbb{N}$. Furthermore, the loss at average-iterate, $\mathbb{E}[\mathcal{R}_T^\pi\left(\frac{1}{K}\sum_{k=0}^{K-1}\Theta(k)\right)]$, admits the same upper bound as the regret upper bound in equation 17, up to a multiplicative factor of 10.*

The proof of Theorem 5.4 can be found in Section B.

Assumption 5.1 is critical to obtain finite-time bounds in Theorem 5.4, and holds when the system can be restarted independently from the initial state distribution Bhandari et al. (2018). In the specific case of fully-observable MDPs, the process $\{(S_k, A_k) : k \in \mathbb{N}\}$ is a Markov chain under any stationary policy, and mixing time arguments under uniform ergodicity assumptions are used for analysis under Markovian sampling from a single trajectory without independent restarts (Bhandari et al., 2018; Cayci et al., 2023). On the other hand, in the case of POMDPs, $\{(S_k, A_k) : k \in \mathbb{N}\}$ is not a Markov chain under a general non-stationary

policy $\pi$. In the specific case of policies parameterized by RNNs with hidden state $\{H_k : k \in \mathbb{N}\}$, the augmented process $\{(S_k, A_k, Y_k, H_k) : k \in \mathbb{N}\}$ forms a Markov process. The challenge here is that the state space for this augmented Markov process may be very large or even continuous, and standard theoretical tools (e.g., mixing time arguments) can become much more involved. Under Assumption 5.3, Theorem 5.4 implies the global $\epsilon$-optimality of Rec-TD as the sequence-length $T \to \infty$ for sufficiently large number of iterations $K = \mathcal{O}(C_T^{(1)}/\epsilon^2)$ and network width $m = \mathcal{O}(C_T^{(2)}/\epsilon^2)$. If we omit Assumption 5.3, the error bound in Theorem 5.4 still holds with an additional error term $\mathcal{O}\left(\frac{1}{1-\gamma}\epsilon_{\text{app}}(\mathscr{F}_T)\right)$ where

$$\epsilon_{\text{app}}(\mathscr{F}_T) := \inf_{f \in \mathscr{F}_T} \mathbb{E}_\mu^\pi \left[ \sum_{t=0}^{T-1} \gamma^t \left( f_t(\bar{Z}_t) - \mathcal{Q}_t^\pi(\bar{Z}_t) \right)^2 \right]$$

is the function approximation error.

*Remark* 5.5 (Overcoming perceptual aliasing with Rec-TD). Memoryless TD learning suffers from a non-vanishing optimality gap in POMDPs, known as perceptual aliasing (Singh et al., 1994). To address this, Rec-TD integrates $T$-step stochastic approximation with an RNN, enabling it to retain memory. Accordingly, Theorem 5.4 establishes that as $T \to \infty$, Rec-TD reduces $\mathcal{R}_\infty^\pi$ to arbitrarily small values, given sufficiently large network width $m$ and iteration count $K$.

*Remark* 5.6 (The impact of long-term dependencies). Note that both constants $C_T^{(1)}, C_T^{(2)}$ polynomially depend on $p_T (\varrho_1 \alpha_m)$. As noted in Goodfellow et al. (2016), the spectral radius of $\{\mathbf{W}(k) : k \in \mathbb{N}\}$ determines the degree of long-term dependencies in the problem as it scales $H_t$. Consistent with this observation, our bounds depend on

$$\alpha_m := \alpha + \frac{\rho_{\mathsf{w}}}{\sqrt{m}} \geq \lambda_{\mathsf{max}}(\mathbf{W}^\top(k)\mathbf{W}(k)) = \max_{i \in [m]} |W_{ii}(k)|,$$

for any $k \in \mathbb{N}$. Note that Theorem 5.4 requires $\rho_{\mathsf{w}} \geq \nu_{\mathsf{w}}$, thus $\max_{i \in [m]} |W_{ii}(k)|$ should be sufficiently large depending on the RKHS norm $\nu$. Let $\varepsilon > 0$ be any given target error.

- **Short-term memory.** If $\alpha_m < \frac{1}{\varrho_1}$, then it is easy to see that $p_T(\varrho_1 \alpha_m) \leq \frac{1}{1-\varrho_1 \alpha_m}$. Thus, the extra term ($\heartsuit$) in equation 17 vanishes at a geometric rate as $T \to \infty$, yet $m$ (network-width) and $K$ (iteration-complexity) are still $\tilde{\mathcal{O}}(1/\varepsilon^2)$. Rec-TD is very efficient in that case.

- **Long-term memory.** If $\alpha_m > \frac{1}{\varrho_1}$, as $T \to \infty$, both $m$ and $K$ grow at a rate $\mathcal{O}\left((\varrho_1 \alpha_m)^T/\varepsilon^2\right)$ while the extra term ($\heartsuit$) in equation 17 vanishes at a geometric rate. As such, the required network size and iterations grow at a geometric rate with $T$ in systems with long-term memory, constituting the pathological case.

Theorem 5.4 emphasizes the critical importance of max-norm projection and large neural network size $m$ in stabilizing the training of IndRNNs by Rec-TD, and guides the choice of the projection radius $\rho$. Interestingly, if $\{\mathcal{Q}_t^\pi : t < T\} \in \mathscr{F}_T$ has an RKHS norm $\nu_{\mathsf{w}} \leq 1/\varrho_1$, then Rec-TD with a projection radius $\rho_{\mathsf{w}} \gtrsim \nu_{\mathsf{w}}$ and overparameterization $m \gg 1$ yields significantly improved policy evaluation performance in terms of $C_T^{(1)}, C_T^{(2)}$ for large $T$. Similar projection mechanisms on $\{W_{ii} : i \in [m]\}$ are widely used for IndRNNs in practice, for instance in Morad et al. (2023), to enhance stability.

The performance of Rec-TD is studied numerically in Random-POMDP instances in Section C.

## 6 Actor: Recurrent Natural Policy Gradient (Rec-NPG) for POMDPs

The goal is to solve the following problem for a given initial distribution $\mu \in \Delta(\mathbb{Y})$ and $\rho \in \mathbb{R}_{>0}^2$:

$$\max_{\Theta \in \mathbb{R}^{m(d+1)}} \mathcal{V}^{\pi^\Phi}(\mu) \text{ such that } \Phi \in \Omega_{\rho,m}, \tag{PO}$$

## 6.1 Recurrent Natural Policy Gradient for POMDPs

In this section, we describe the recurrent natural policy gradient (Rec-NPG) algorithm for non-stationary reinforcement learning. First, we formally establish in Prop. D.2 that the policy gradient under partial observability takes the form

$$\nabla_\Phi \mathcal{V}^{\pi^\Phi}(\mu) := \mathbb{E}_\mu^{\pi^\Phi}\left[\sum_{t=0}^\infty \gamma^t \mathcal{Q}_t^{\pi^\Phi}(Z_t, A_t)\nabla_\Phi \ln \pi_t^\Phi(A_t|Z_t)\right],$$

where the state $S_t$ in the MDP framework is replaced by the process history $Z_t$ in POMDP. Fisher information matrix under a policy $\pi^\Phi$ is defined as

$$G_\mu(\Phi) := \mathbb{E}_\mu^{\pi^\Phi}\left[\sum_{t=0}^\infty \gamma^t \nabla \ln \pi_t^\Phi(A_t|Z_t)\nabla^\top \ln \pi_t^\Phi(A_t|Z_t)\right],$$

for an initial observation distribution $\mu \in \Delta(\mathbb{Y})$. Rec-NPG updates the policy parameters by

$$\Phi(n+1) = \Phi(n) + \eta \cdot G_\mu^\dagger(\Phi(n))\nabla_\Phi \mathcal{V}^{\pi^{\Phi(n)}}(\mu), \tag{18}$$

for an initial parameter $\Phi(0)$ and step-size $\eta > 0$, where $G^\dagger$ denotes the Moore-Penrose inverse of a matrix $G$. This update rule is in the same spirit as the NPG introduced in Kakade (2001), however, due to the non-stationary nature of the partially observable MDP, it has significant complications that we will address.

In order to avoid computationally-expensive policy updates in equation 18, we utilize the following extension of the compatible function approximation in Kakade (2001) to the case of non-stationary policies for POMDPs.

**Proposition 6.1** (Compatible function approximation for non-stationary policies). *For any $\Phi \in \mathbb{R}^{m(d+1)}$ and initial observation distribution $\mu$, let*

$$\mathcal{L}_\mu(w; \Phi) = \mathbb{E}_\mu^{\pi^\Phi}\left[\sum_{t=0}^\infty \gamma^t \left(\nabla^\top \ln \pi_t^\Phi(A_t|Z_t)\omega - \mathcal{A}_t^{\pi^\Phi}(\bar{Z}_t)\right)^2\right], \tag{19}$$

*for $\omega \in \mathbb{R}^{m(d+1)}$. Then, we have*

$$G_\mu^\dagger(\Phi)\nabla_\Phi \mathcal{V}^{\pi^\Phi}(\mu) \in \underset{\omega\in\mathbb{R}^{m(d+1)}}{\arg\min}\, \mathcal{L}_\mu(\omega; \Phi). \tag{20}$$

We have the following remark regarding the intricacies of compatible function approximation in the POMDP setting.

*Remark* 6.2 (Path-based compatible function approximation with truncation). For MDPs, the compatible function approximation error $\mathcal{L}_\mu(w; \Phi)$ can be expressed by using the discounted state-action occupancy measure, from which one can obtain unbiased samples (Agarwal et al., 2020; Konda & Tsitsiklis, 2003). Thus, the infinite-horizon can be handled without any loss. On the other hand, for POMDPs as in equation 19, this simplification is impossible due to the non-stationarity. As such, we use a path-based method for a sequence-length $T \in \mathbb{N}$ with

$$\ell_T(\omega; \Phi, \mathcal{Q}) := \sum_{t=0}^{T-1} \gamma^t (\nabla \ln \pi_t^\Phi(A_t|Z_t)\omega - \mathcal{A}_t(Z_t, A_t))^2,$$

where $\mathcal{A}_t(z_t, a_t) = \mathcal{Q}_t(z_t, a_t) - \sum_{a\in\mathbb{A}} \pi_t^\Phi(a|z_t)\mathcal{Q}_t(z_t, a)$ is the advantage function.

Given a policy with parameter $\Phi(n)$, the corresponding output of the critic, which is obtained by Rec-TD with the average-iterate as

$$\hat{\mathcal{Q}}^{(n)}(\cdot) := F_t(\cdot; \bar{\Theta}^n) \text{ for } \bar{\Theta}^n := \frac{1}{K_{\text{td}}}\sum_{k<K_{\text{td}}} \Theta^n(k),$$

the actor aims to solve the following problem:

$$\min_{\omega \in \Omega_{\rho,m}} \mathbb{E}\left[\ell_T\left(\omega; \Phi(n), \hat{\mathcal{Q}}^{(n)}\right) \Big| \bar{\Theta}^n, \Phi(n), \ldots, \Phi(0)\right].$$

We utilize stochastic gradient descent (SGD) to solve the above problem. Let $\bar{Z}_T^{n,k} \sim P_T^{\pi^{\Phi(n)},\mu}$ be an independent random sequence for $k \in \mathbb{N}$, $\hat{\omega}_n(0) = 0$, and

$$\tilde{\omega}_n(k+1) = \hat{\omega}_n(k) - \eta_{\mathsf{sgd}} \nabla_\omega \ell_T\left(\hat{\omega}_n(k); \Phi(n), \hat{\mathcal{Q}}^{(n)}\right),$$
$$\hat{\omega}_n(k+1) = \mathbf{Proj}_{\Omega_{\rho,m}}[\tilde{\omega}_n(k+1)],$$

A stochastic estimate of $G_\mu^\dagger(\Phi(n))\nabla_\Phi \mathcal{V}^{\pi^{\Phi(n)}}(\mu)$ is computed as $\omega_n := \frac{1}{K_{\mathsf{sgd}}}\sum_{k<K_{\mathsf{sgd}}} \hat{\omega}_n(k)$, followed by

$$\Phi(n+1) = \Phi(n) + \eta_{\mathsf{npg}} \cdot \omega_n.$$

In the following, we present a theoretical analysis of this policy optimization algorithm.

## 6.2 Theoretical Analysis of Rec-NAC for POMDPs

We establish an error bound on the best-iterate for the Rec-NPG. The significance of the following result is two-fold: (i) it will explicitly connect the optimality gap to the compatible function approximation error, and (ii) it will explicitly show the impact of truncation on the performance of path-based policy optimization for the non-stationary case.

**Theorem 6.3.** *Assume that $P_T^{\pi^\star,\mu}$ is absolutely continuous with respect to $P_T^{\pi^{\Phi(n)},\mu}$ for all $n < N$. Under this assumption, let*

$$\kappa := \max_{0 \le n < N} \left\|\frac{P_T^{\pi^\star,\mu}}{P_T^{\pi^{\Phi(n)},\mu}}\right\|_\infty$$

*be the concentrability coefficient, and*

$$V_n := \mathcal{V}^{\pi^\star}(\mu) - \mathcal{V}^{\pi^{\Phi(n)}}(\mu), \ n < N$$

*be the optimality gap. Rec-NPG after $N \in \mathbb{Z}_+$ steps with step-size $\eta_{\mathsf{npg}} = \frac{1}{\sqrt{N}}$ and projection radius $\rho \in \mathbb{R}_{>0}^2$ yields*

$$\min_{0 \le n < N} \mathbb{E}_0[V_n] \lesssim \frac{\ln|\mathbb{A}|}{(1-\gamma)\sqrt{N}} + \frac{\|\rho\|_2^2}{1-\gamma}\frac{p_T(\alpha_m \varrho_1)}{m^{\frac{1}{4}}} + \frac{\gamma^T r_\infty}{(1-\gamma)^2} + \frac{\sqrt{\kappa}}{N\sqrt{1-\gamma}}\sum_{n=0}^{N-1} \mathbb{E}_0\left(\varepsilon_{\mathsf{cfa}}^T(\Phi(n), \omega_n)\right)^{\frac{1}{2}},$$

*where $\mathbb{E}_0$ is the conditional expectation given the symmetric random initialization $(c^0, \Phi(0)) \sim \zeta_0$, and*

$$\varepsilon_{\mathsf{cfa}}^T(\Phi, \omega) := \sum_{t<T} \gamma^t |\nabla^\top \ln \pi_t^\Phi(A_t|Z_t)\omega - \mathcal{A}_t^{\pi^\Phi}(Z_t, A_t)|^2.$$

*Remark* 6.4. We have the following remarks.

- The effectiveness of Rec-NPG is proportional to the approximation power of the IndRNN used for policy parameterization, as reflected in $\varepsilon_{\mathsf{cfa}}^T$ in Theorem 6.3. We further characterize this error term in Propositions 6.6-6.8 in the following.

- The terms $L_t, \beta_t, \Lambda_t, \chi_t$ grow at a rate $p_t(\varrho_1 \alpha_m)$. Thus, if $\alpha_m > \varrho_1^{-1}$, then $m$ and $N$ should grow at a rate $(\alpha_m \varrho_1)^T$, implying the curse of dimensionality (more generally, it is known as the exploding gradient problem Goodfellow et al. (2016)). On the other hand, if $\alpha_m < \varrho_1^{-1}$, then $L_t, \beta_t, \Lambda_t, \chi_t$ are all $\mathcal{O}(1)$ for all $t$, implying efficient learning of POMDPs. This establishes a very interesting connection between the memory in the system, the continuity and smoothness of the RNN with respect to its parameters, and the optimality gap under Rec-NPG.

- The term $\frac{2\gamma^T r_\infty}{(1-\gamma)^2}$ is due to truncating the trajectory at $T$, and vanishes with large $T$.

- Rec-NPG achieves $\epsilon$-optimality (up to the compatible function approximation and truncation errors) with $N = \mathcal{O}(1/\epsilon^2)$ steps and $m = \mathcal{O}(1/\epsilon^4)$ neural network width for any $\epsilon > 0$.

*Remark* 6.5. The quantity $\kappa$ in Proposition 6.8 is the so-called concentrability coefficient in policy gradient methods (Agarwal et al., 2020; Bhandari & Russo, 2019; Wang et al., 2019), and determines the complexity of exploration. Note that it is defined in terms of path probabilities $P_T^{\pi,\mu}$ in the non-stationary setting. By making the assumption $\kappa < \infty$, we assume that the policies $\pi^{\Phi(n)}$ perform sufficient exploration to visit each trajectory visited by $\pi^\star$ with positive probability. In order to establish similar bounds without this assumption, entropic regularization is widely used to encourage exploration in practical scenarios Ahmed et al. (2019); Cen et al. (2020); Cayci et al. (2024c). The benefits of using entropic regularization in policy optimization for POMDPs to encourage exploration is an interesting future research direction.

In the following, we decompose the compatible function approximation error $\varepsilon_{\mathsf{cfa}}^T$ into the approximation error for the RNN and the statistical errors. To that end, let

$$\varepsilon_{\mathsf{app},n} = \inf_{\omega \in \Omega_{\rho,m}} \mathbb{E} \sum_{t<T} \gamma^t \big| \nabla^\top F_t(\bar{Z}_t; \Phi(0))\omega - Q_t^{\pi^{\Phi(n)}}(\bar{Z}_t) \big|^2,$$

be the approximation error where the expectation is with respect to $P_T^{\pi^{\Phi(n)},\mu}$,

$$\varepsilon_{\mathsf{td},n} = \mathbb{E}[\mathcal{R}_T^{\pi^{\Phi(n)}}(\bar{\Theta}^{(n)})|\Phi(k), k \leq n],$$

be the error in the critic (see equation 16), and finally let

$$\varepsilon_{\mathsf{sgd},n} = \mathbb{E}[\ell_T(\omega_n; \Phi(n), \hat{\mathcal{Q}}^{(n)})|\bar{\Theta}^{(n)}, \Phi(k), k \leq n] - \inf_w \mathbb{E}[\ell_T(\omega; \Phi(n), \hat{\mathcal{Q}}^{(n)})|\bar{\Theta}^{(n)}, \Phi(k), k \leq n],$$

be the error in the policy update via compatible function approximation.

**Proposition 6.6** (Error decomposition for $\varepsilon_{\mathsf{cfa}}^T$). *For any $n \in \mathbb{Z}_+$, we have*

$$\mathbb{E}\Big[\mathbb{E}_\mu^{\pi^{\Phi(n)}}\big[\ell_T(\omega_n; \Phi(n), \mathcal{Q}^{(n)})\big]\,\Big|\,\Phi(k), k \leq n\Big] \leq \frac{8\|\rho\|_2^2}{m} \sum_{t=0}^{T-1} \gamma^t \beta_t^2 + 8\varepsilon_{\mathsf{app},n} + 6\varepsilon_{\mathsf{td},n} + 2\varepsilon_{\mathsf{sgd},n}.$$

From Theorem 5.4, we have, for $\eta_{\mathsf{td}} = \mathcal{O}(1/\sqrt{K_{\mathsf{td}}})$,

$$\varepsilon_{\mathsf{td},n} \leq \mathbf{poly}(p_T(\varrho_1 \alpha_m))\mathcal{O}\left(\frac{1}{\sqrt{K_{\mathsf{td}}}} + \frac{1}{\sqrt{m_{\mathsf{critic}}}} + \gamma^T\right),$$

and by Theorem 14.8 in Shalev-Shwartz & Ben-David (2014), we have, for $\eta_{\mathsf{sgd}} = \mathcal{O}(1/\sqrt{K_{\mathsf{sgd}}})$,

$$\varepsilon_{\mathsf{sgd},n} \leq \mathbf{poly}(p_T(\varrho_1 \alpha_m), \|\rho\|_2)\mathcal{O}(1/\sqrt{K_{\mathsf{sgd}}}).$$

As such, the statistical errors in the critic and the policy update (i.e., $\varepsilon_{\mathsf{td},n}, \varepsilon_{\mathsf{sgd},n}$) can be made arbitrarily small by using larger $K_{\mathsf{td}}, K_{\mathsf{sgd}}$ and larger $m_{\mathsf{critic}}$. The remaining quantity to characterize is the approximation error, which is of critical importance for a small optimality gap as shown in Theorem 6.3 and Proposition 6.6. In the following, we will provide a finer characterization of $\varepsilon_{\mathsf{app},n}$ and identify a class of POMDPs that can be efficiently solved using Rec-NPG.

**Assumption 6.7.** For an index set $J$ and $\nu \in \mathbb{R}_{>0}^2$, we consider a class $\mathscr{H}_{J,\nu}$ of transportation mappings

$$\left\{ \boldsymbol{v}^{(j)} \in \mathscr{H} : j \in J, \begin{pmatrix} \sup_{\theta \in \mathbb{R}^{d+1}, j \in J} |v_w^{(j)}(\theta)| \\ \sup_{\theta \in \mathbb{R}^{d+1}, j \in J} \|v_u^{(j)}(\theta)\|_2 \end{pmatrix} \leq \begin{pmatrix} \nu_w \\ \nu_u \end{pmatrix} \right\},$$

and also the corresponding infinite-width limit

$$\mathscr{F}_{J,\nu} := \{\bar{z} \mapsto \mathbb{E}[\Psi(\bar{z};\theta_0)\boldsymbol{v}(\theta_0)] : \boldsymbol{v} \in \mathbf{Conv}(\mathscr{H}_{J,\nu})\},$$

where $\Psi(\cdot;\theta_0)$ is the NTRF matrix, defined in equation 12.

We assume that there exists an index set $J$ and $\nu \in \mathbb{R}^2_{>0}$ such that $\mathcal{Q}^{\pi^{\Phi(n)}} \in \mathscr{F}_{J,\nu}$ for all $n \in \mathbb{N}$.

This representational assumption implies that the $\mathcal{Q}$-functions under all iterate policies $\pi^{\Phi(n)}$ throughout the Rec-NPG iterations $n = 0, 1, \dots$ can be represented by convex combinations of a *fixed* set of mappings in the NTK function class $\mathscr{F}$ indexed by $J$. As we will see, the richness of $J$ as measured by a relevant Rademacher complexity will play an important role in bounding the approximation error. To that end, for $\bar{z}_t = (z_t, a_t) \in (\mathbb{Y} \times \mathbb{A})^{t+1}$, let

$$G_t^{\bar{z}_t} := \{\phi \mapsto \nabla_\phi^\top H_t^{(1)}(\bar{z}_t;\phi)\boldsymbol{v}(\phi) : \boldsymbol{v} \in \mathscr{H}_{J,\nu}\},$$

and

$$\mathrm{Rad}_m(G_t^{\bar{z}_t}) := \mathop{\mathbb{E}}_{\substack{\epsilon \sim \mathsf{Rad}^m(1) \\ \Phi(0) \sim \zeta_{\mathsf{init}}}} \sup_{g \in G_t^{\bar{z}_t}} \frac{1}{m} \sum_{i=1}^m \epsilon_i g(\Phi_i(0)).$$

Note that $\boldsymbol{v} \in \mathscr{H}_{J,\nu}$ above can be replaced with $\boldsymbol{v} \in \mathbf{Conv}(\mathscr{H}_{J,\nu})$ without any loss. In that case, since the mapping $\boldsymbol{v}^{(j)} \mapsto f_t^\star(\bar{z}_t;\boldsymbol{v}^{(j)}) \in G_t^{\bar{z}_t}$ is linear, $G_t^{\bar{z}_t}$ is replaced with $\mathbf{Conv}(G_t^{\bar{z}_t})$ without changing the Rademacher complexity (Mohri et al., 2018).

The following provides a finer characterization of the approximation error.

**Proposition 6.8.** *Under Assumption 6.7, if $\rho \succeq \nu$, then*

$$\epsilon_{\mathsf{app},n} \leq \frac{1}{1-\gamma}\left(2 \max_{0 \leq t < T} \max_{\bar{z}_t \in (\mathbb{Y} \times \mathbb{A})^{t+1}} \mathrm{Rad}_m(G_t^{\bar{z}_t}) + L_T \|\rho\|_2 \sqrt{\frac{\ln\left(2T|\mathbb{Y} \times \mathbb{A}|^T/\delta\right)}{m}}\right)^2,$$

*for all $n$ simultaneously with probability at least $1 - \delta$ over the random initialization for any $\delta \in (0, 1)$.*

*Remark* 6.9. An interesting case that lead to a vanishing approximation error (as $m \to \infty$) is $|J| < \infty$. Then, Proposition 6.8 reduces to Cayci et al. (2024b) (with $T = 1$ for FNNs) with the complexity term $\mathcal{O}\left(\sqrt{\frac{\ln(|J|/\delta)}{m}}\right)$ by the finite-class lemma (Mohri et al., 2018). In this case, the $\mathcal{Q}$-functions throughout $n = 0, 1, \dots$ lie in the convex hull of $|J|$ fixed functions in $\mathscr{F}$ generated by $\{\boldsymbol{v}^{(j)} \in \mathscr{H} : j \in J\}$.

*Remark* 6.10. As noted in Cayci et al. (2024b), in a *static* problem (e.g., the regression problem in supervised learning or policy evaluation in Section 5) with a target function $f \in \mathscr{F}$, the approximation error is easy to characterize:

$$\left|\nabla^\top F_t(\bar{z}_t; \Phi(0))\omega^\star - f_t(\bar{z}_t)\right| = \mathcal{O}\left(\sqrt{\frac{\ln(1/\delta)}{m}}\right), \tag{21}$$

by Hoeffding inequality with $\omega^\star := \left[\frac{1}{\sqrt{m}}c_i\boldsymbol{v}(\Phi_i(0))\right]_{i \in [m]}$.

In the *dynamical* policy optimization problem, the representational assumption $\mathcal{Q}^{\pi^{\Phi(n)}} \in \mathscr{F}$ does not imply arbitrarily small approximation error as $m \to \infty$ since the function $\mathcal{Q}^{\pi^{\Phi(n)}}$ also depends on $\Phi(0)$. Thus,

$$\nabla^\top F_t(\bar{z}_t; \Phi(0))\omega_n^\star = \sum_{i=1}^m \frac{\nabla^\top H_t^{(i)}(\bar{z}_t; \Phi(0))\boldsymbol{v}^{\Phi(n)}(\Phi_i(0))}{m}$$

with $\omega_n^\star := [\frac{1}{\sqrt{m}}c_i\boldsymbol{v}^{\Phi(n)}(\Phi_i(0))]_{i \in [m]}$ for $\boldsymbol{v}^{\Phi(n)} \in \mathscr{H}$ may not converge to the target function $\mathcal{Q}^{\pi^{\Phi(n)}}$ as $m \to \infty$ because of the correlated $\nabla^\top H_t^{(i)}(\bar{z}_t; \Phi(0))\boldsymbol{v}^{\Phi(n)}(\Phi_i(0))$ across $i \in [m]$. To address this, we characterize the uniform approximation error as in Proposition 6.8 for the random features of the actor RNN in approximating all $\mathcal{Q}^{\pi^{\Phi(n)}}$ for all $n$ based on Rademacher complexity.

# 7 Conclusion

We studied RNN-based policy evaluation and policy optimization methods with finite-time analyses, which demonstrate the effectiveness of the NPG method equipped with RNNs for POMDPs. An important limitation of Rec-NPG is that its memory and sample complexity significantly increases in POMDPs with long-term dependencies as pointed out in Remarks 5.6-6.4. In order to mitigate these issues, as an extension of this work, input normalization (Zucchet & Orvieto, 2024) and preconditioned Rec-TD updates to incorporate curvature information (Martens & Sutskever, 2011) are important directions for future research.

### Acknowledgments

This work was funded by the Federal Ministry of Education and Research (BMBF) and the Ministry of Culture and Science of the German State of North Rhine-Westphalia (MKW) under the Excellence Strategy of the Federal Government and the Länder. Atilla Eryilmaz's research was supported in part by NSF AI Institute (AI-EDGE) 2112471, CNS-NeTS-2106679; and the ARO Grant W911NF-24-1-0103.

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

# A   Algorithmic Tools for Recurrent Neural Networks

## A.1   Max-Norm Projection for Recurrent Neural Networks

Max-norm regularization, proposed by Srebro et al. (2004), has been shown to be very effective across a broad spectrum of deep learning problems (Srivastava et al., 2014; Goodfellow et al., 2013). In this work, we incorporate max-norm regularization (around the random initialization) into the recurrent natural policy gradient for sharp convergence guarantees. To that end, given an initialization $(\mathbf{W}(0), \mathbf{U}(0), c)$ as in Definition 3.1 and a vector $\rho = (\rho_\mathsf{w}, \rho_\mathsf{u})^\top \in \mathbb{R}^2_{>0}$ of projection radii, we define the compactly-supported set of weights $\Omega_{\rho,m} \subset \mathbb{R}^{m(d+1)}$ as

$$\Omega_{\rho,m} = \left\{ \Theta \in \mathbb{R}^{m(d+1)} : \max_i |W_{ii} - W_{ii}(0)| \leq \frac{\rho_\mathsf{w}}{\sqrt{m}}, \ \max_i \|U_i - U_i(0)\| \leq \frac{\rho_\mathsf{u}}{\sqrt{m}} \right\}. \tag{22}$$

Given any symmetric random initialization $(\mathbf{W}(0), \mathbf{U}(0), c)$ and $\rho \in \mathbb{R}^2_{>0}$, the set $\Omega_{\rho,m}$ is a compact and convex subset of $\mathbb{R}^{m(d+1)}$, and for any $\Theta \in \Omega_{\rho,m}$, we have

$$\max_{1 \leq i \leq m} |W_{ii} - W_{ii}(0)| \leq \frac{\rho_\mathsf{w}}{\sqrt{m}},$$

$$\max_{1 \leq i \leq m} \|U_i - U_i(0)\| \leq \frac{\rho_\mathsf{u}}{\sqrt{m}}.$$

Let

$$\mathbf{Proj}_{\Omega_{\rho,m}}[\Theta] = \left[ \operatorname*{arg\,min}_{w \in \mathcal{B}_2\left(W_{ii}(0), \frac{\rho_\mathsf{w}}{\sqrt{m}}\right)} |W_{ii} - w_i|, \ \operatorname*{arg\,min}_{u_i \in \mathcal{B}_2\left(U_i(0), \frac{\rho_\mathsf{u}}{\sqrt{m}}\right)} \|\mathbf{U}_i - u_i\|_2 \right]_{i \in [m]} \tag{23}$$

As such, the projection operator $\mathbf{Proj}_{\Omega_{\rho,m}}[\cdot]$ onto $\Omega_{\rho,m}$ is called the max-norm projection (or regularization).

Note that we have $\|\mathbf{W} - \mathbf{W}(0)\|_2 \leq \rho_\mathsf{w}$, $\|\mathbf{U} - \mathbf{U}(0)\|_2 \leq \rho_\mathsf{u}$ and $\|\Theta - \Theta(0)\|_2 \leq \|\rho\|_2$ in the $\ell_2$ geometry for any $\Theta \in \Omega_{\rho,m}$. Therefore, although the max-norm parameter class $\Omega_{\rho,m} \subset \{\Theta \in \mathbb{R}^{m(d+1)} : \|\Theta - \Theta(0)\|_2 \leq \|\rho\|_2\}$, the $\ell_2$-projected Cai et al. (2019); Wang et al. (2019); Liu et al. (2019) and max-norm projected Cayci et al. (2024b) optimization algorithms recover exactly the same function class (i.e., RKHS associated with the neural tangent kernel studied in Ji et al. (2019); Telgarsky (2021), see Section 3.1).

# B   Proofs for Section 5

An important quantity in the analysis of recurrent neural networks is the following:

$$\Gamma_t^{(i)}(\bar{z}_t; \Theta) := W_{ii} H_t^{(i)}(\bar{z}_t; \Theta),$$

for any hidden unit $i \in [m]$ and $\Theta \in \mathbb{R}^{m(d+1)}$. The following Lipschitzness and smoothness results for $\Theta_i \mapsto H_t^{(i)}(\bar{z}_t; \Theta)$ and $\Theta_i \mapsto \Gamma_t^{(i)}(\bar{z}_t; \Theta)$.

**Lemma B.1** (Local continuity of hidden states; Lemma 1-2 in Cayci & Eryilmaz (2025)). *Given $\rho \in \mathbb{R}_{>0}^2$ and $\alpha \geq 0$, let $\alpha_m = \alpha + \frac{\rho_w}{\sqrt{m}}$. Then, for any $\bar{\boldsymbol{z}} \in (\mathbb{Y} \times \mathbb{A})^{\bar{Z}_+}$ with $\sup_{t \in \mathbb{N}} \left\| \begin{pmatrix} y_t \\ a_t \end{pmatrix} \right\|_2 \leq 1$, $t \in \mathbb{N}$ and $i \in [m]$,*

- *$\Theta_i \mapsto H_t^{(i)}(\bar{z}_t; \Theta)$ is $L_t$-Lipschitz continuous with $L_t = (\varrho_0^2 + 1)\varrho_1^2 \cdot p_t^2(\alpha_m \varrho_1)$,*

- *$\Theta_i \mapsto H_t^{(i)}(\bar{z}_t; \Theta)$ is $\beta_t$-smooth with $\beta_t = \mathcal{O}\left(d \cdot p_t(\alpha_m \varrho_1) \cdot q_t(\alpha_m \varrho_1)\right)$,*

- *$\Theta_i \mapsto \Gamma_t^{(i)}(\bar{z}_t; \Theta)$ is $\Lambda_t$-Lipschitz with $\Lambda_t = \sqrt{2}(\varrho_0 + 1 + \alpha_m L_t)$,*

- *$\Theta_i \mapsto \Gamma_t^{(i)}(\bar{z}_t; \Theta)$ is $\chi_t$-smooth with $\chi_t = \sqrt{2}(L_t + \alpha_m \beta_t)$,*

*in $\Omega_{\rho,m}$. Consequently, for any $\Theta \in \Omega_{\rho,m}$,*

$$\sup_{\bar{\boldsymbol{z}} \in \bar{\mathbb{H}}_\infty} \max_{0 \leq t \leq T} |F_t(\bar{z}_t; \Theta)| \leq L_T \cdot \|\rho\|_2, \ T \in \mathbb{N}, \tag{24}$$

$$\sup_{\bar{\boldsymbol{z}} \in \bar{\mathbb{H}}_\infty} |F_t^{\mathsf{Lin}}(\bar{z}_t; \Theta) - F_t(\bar{z}_t; \Theta)| \leq \frac{2}{\sqrt{m}}(\varrho_2 \Lambda_t^2 + \varrho_1 \chi_t)\|\Theta - \Theta(0)\|_2^2, \ t \in \mathbb{N}, \tag{25}$$

$$\sup_{\bar{\boldsymbol{z}} \in \bar{\mathbb{H}}_\infty} \left\langle \nabla F_t(\bar{z}_t; \Theta) - \nabla F_t(\bar{z}_t; \Theta(0)), \Theta - \bar{\Theta} \right\rangle \leq \frac{2\beta_t^2 \|\rho\|_2^2}{\sqrt{m}}, \tag{26}$$

*with probability 1 over the symmetric random initialization $(\mathbf{W}(0), \mathbf{U}(0), c) \sim \zeta_0$.*

The following result builds on Proposition 3.8 in Cayci & Eryilmaz (2025), and identifies the approximation error for approximating $f^\star \in \mathscr{F}$ by using randomly-initialized IndRNNs of width $m$. Unlike the supervised learning setting in Cayci & Eryilmaz (2025), the approximation error in the RL setting is $P_T^{\mu,\pi}$-norm.

**Lemma B.2** (Approximation error between RNN-NTRF and RNN-NTK). *Let $f^\star \in \mathscr{F}$ with the transportation mapping $\boldsymbol{v} \in \mathscr{H}$, and let*

$$\bar{\Theta}_i = \Theta_i(0) + \frac{1}{\sqrt{m}} c_i \boldsymbol{v}(\Theta_i(0)), i \in [m]. \tag{27}$$

*for the initialization $(\mathbf{W}(0), \mathbf{U}(0), c) \sim \zeta_0$ in Def. 3.1. Let*

$$F_t^{\mathsf{Lin}}(\cdot; \Theta) = \nabla_\Theta F_t(\cdot; \Theta(0)) \cdot (\Theta - \Theta(0)).$$

*If $P_T^{\pi,\mu}$ induces a compactly-supported marginal distribution for $X_t, t \in \mathbb{N}$ such that $\|X_t\|_2 \leq 1$ a.s. and $\{\bar{Z}_t : t \in \mathbb{N}\}$ is independent from the random initialization $(\mathbf{W}(0), \mathbf{U}(0), c)$, then we have*

$$\mathbb{E}\left[\mathbb{E}_\mu^\pi\left[\left(f_t^\star(\bar{Z}_t) - F_t^{\mathsf{Lin}}(\bar{Z}_t; \bar{\Theta})\right)^2\right]\right] \leq \frac{2\|\nu\|_2^2(1 + \varrho_0^2)p_t^2(\alpha \varrho_1)}{m}, \tag{28}$$

*where the outer expectation is with respect to the random initialization $(\mathbf{W}(0), \mathbf{U}(0), c) \sim \zeta_0$.*

*Proof.* For any hidden unit $i \in [m]$, let

$$\zeta_i = \left\langle \boldsymbol{v}(\Theta_i(0)), \sum_{k=0}^t W_{ii}{}^k(0) \begin{pmatrix} H_{t-k-1}^{(i)}(\bar{Z}_{t-k-1}, \Theta_i(0)) \\ X_{t-k} \end{pmatrix} \prod_{j=0}^k \mathcal{I}_{t-j}(\bar{Z}_{t-j}; \Theta_i(0)) \right\rangle.$$

Then, it is straightforward to see that

$$F_t^{\mathsf{Lin}}(\bar{Z}_t; \bar{\Theta}) = \frac{1}{m} \sum_{i=1}^m \zeta_i, \tag{29}$$

and $\mathbb{E}[\zeta_i|\bar{Z}_t] = \mathbb{E}[f_t^\star(\bar{Z}_t)|\bar{Z}_t]$ almost surely. Note that $\{\zeta_i : i \in [m/2]\}$ is independent and identically distributed and $\zeta_i = \zeta_{i+m/2}$ for any $i \in [m/2]$. Also, with probability 1 we have

$$
\begin{aligned}
|\zeta_i| &\overset{(\spadesuit)}{\leq} \|\boldsymbol{v}(\Theta_i(0))\|_2 \cdot \left\| \sum_{k=0}^{t} W_{ii}{}^k(0) \begin{pmatrix} H_{t-k-1}^{(i)}(\bar{Z}_{t-k-1}, \Theta_i(0)) \\ X_{t-k} \end{pmatrix} \prod_{j=0}^{k} \mathcal{I}_{t-j}(\bar{Z}_{t-j}; \Theta_i(0)) \right\|_2, \\
&\overset{(\clubsuit)}{\leq} \|\boldsymbol{v}(\Theta_i(0))\|_2 \sum_{k=0}^{t-1} \alpha^k \varrho_1^{k+1} \sqrt{1+\varrho_0^2}, \\
&\overset{(\diamondsuit)}{\leq} \|\nu\|_2 \cdot \varrho_1 \cdot \sqrt{1+\varrho_0^2} \cdot p_t(\alpha\varrho_1),
\end{aligned}
$$

where ($\spadesuit$) follows from Cauchy-Schwarz inequality, ($\clubsuit$) follows from the uniform bound $\sup_{z\in\mathbb{R}}|\varrho(z)| \leq \varrho_1$ and almost-sure bounds $\|X_k\|_2 \leq 1$ and $|W_{ii}(0)| \leq \alpha$, and ($\diamondsuit$) follows from $\boldsymbol{v} \in \mathscr{H}_\nu$. From these bounds,

$$
\mathrm{Var}(\zeta_i) \leq \mathbb{E}[\mathbb{E}_\mu^\pi[|\zeta_i|^2]] \leq \|\nu\|_2^2 \varrho_1^2 (1+\varrho_0)^2 p_t^2(\alpha\varrho_1), \ i \in [m]. \tag{30}
$$

Therefore,

$$
\begin{aligned}
\mathbb{E}\left[ \mathbb{E}_\mu^\pi \left[ \left( f_t^\star(\bar{Z}_t) - F_t^{\mathsf{Lin}}(\bar{Z}_t; \bar{\Theta}) \right)^2 \right] \right] &= \mathbb{E}_\mu^\pi \left[ \mathbb{E}\left[ \left| \frac{1}{m} \sum_{i=1}^{m} \left( \zeta_i - \mathbb{E}[\zeta_i|\bar{Z}_t] \right) \right|^2 \right] \right], \\
&= \mathbb{E}_\mu^\pi \left[ \mathbb{E}\left[ \left| \frac{2}{m} \sum_{i=1}^{m/2} \left( \zeta_i - \mathbb{E}[\zeta_i|\bar{Z}_t] \right) \right|^2 \right] \right], \\
&= \frac{4}{m^2} \mathbb{E}_\mu^\pi \sum_{i=1}^{m/2} \sum_{j=1}^{m/2} \mathbb{E}\left[ \left( \zeta_i - \mathbb{E}[\zeta_i|\bar{Z}_t] \right) \left( \zeta_j - \mathbb{E}[\zeta_j|\bar{Z}_t] \right) \right], \\
&= \frac{4}{m^2} \mathbb{E}_\mu^\pi \sum_{i=1}^{m/2} \mathrm{Var}(\zeta_i) \leq \frac{2}{m} \|\nu\|_2^2 \varrho_1^2 (1+\varrho_0)^2 p_t^2(\alpha\varrho_1),
\end{aligned}
$$

where the first identity is from Fubini's theorem, the second identity is from the symmetricity of the random initialization, the fourth identity is due to the independent initialization for $i \leq m/2$, and the inequality is from the bound in equation 30.

$\square$

**Proposition B.3** (Non-stationary Bellman equation). *For $\pi \in \Pi_{\mathsf{NM}}$, we have*

$$
\mathcal{Q}_t^\pi(\bar{z}_t) = \mathbb{E}^\pi\left[ r(S_t, A_t) + \gamma \mathcal{Q}_{t+1}^\pi(\bar{Z}_{t+1}) \Big| \bar{Z}_t = \bar{z}_t \right] = \mathbb{E}^\pi\left[ r(S_t, A_t) + \gamma \mathcal{V}_{t+1}^\pi(Z_{t+1}) \Big| \bar{Z}_t = \bar{z}_t \right],
$$

*for any $t \in \mathbb{Z}_+$.*

*Proof of Theorem 5.4.* Since $\{\mathcal{Q}_t^\pi : t \in \mathbb{N}\} \in \mathscr{F}$, let the point of attraction $\bar{\Theta}$ be defined as in equation 27, and the potential function be defined as

$$
\Psi(\Theta) = \|\Theta - \bar{\Theta}\|_2^2. \tag{31}
$$

Then, from the non-expansivity of the projection operator onto the convex set $\Omega_{\rho,m}$, we have the following inequality:

$$
\Psi(\Theta(k+1)) \leq \Psi(\Theta(k)) + 2\eta \sum_{t=0}^{T-1} \gamma^t \delta_t(\bar{Z}_{t+1}^k; \Theta(k)) \left\langle \nabla F_t(\bar{Z}_t^k; \Theta(k)), \Theta(k) - \bar{\Theta} \right\rangle + \eta^2 \|\check{\nabla}\mathcal{R}_T(\bar{Z}_T^k; \Theta(k))\|_2^2. \tag{32}
$$

Let $\check{\mathbb{E}}_t^k[\cdot] := \mathbb{E}[\cdot|\Theta(k), \dots, \Theta(0), \bar{Z}_t^k]$. Then, we obtain

$$\mathbb{E}[\Psi(\Theta(k+1)) - \Psi(\Theta(k))] \le 2\eta\mathbb{E}\Big[\sum_{t=0}^{T-1} \gamma^t \underbrace{\check{\mathbb{E}}_t^k[\delta_t(\bar{Z}_{t+1}^k; \Theta(k))]\left\langle \nabla F_t(\bar{Z}_t^k; \Theta(k)), \Theta(k) - \bar{\Theta}\right\rangle}_{(\spadesuit)_t}\Big]$$
$$+ \eta^2\mathbb{E}\underbrace{\|\check{\nabla}\mathcal{R}_T(\bar{Z}_T^k; \Theta(k))\|_2^2}_{(\clubsuit)}. \quad (33)$$

**Bounding $\mathbb{E}(\spadesuit)_t$.** By using the Bellman equation in the non-stationary setting (cf. Proposition B.3), notice that

$$\check{\mathbb{E}}_t^k \delta_t(\bar{Z}_{t+1}^k; \Theta(k)) = \check{\mathbb{E}}_t^k[r_t^k + \gamma F_{t+1}(\bar{Z}_{t+1}^k; \Theta(k)] - F_t(\bar{Z}_t^k; \Theta(k)),$$
$$= \gamma\check{\mathbb{E}}_t^k\left[F_{t+1}(\bar{Z}_{t+1}^k; \Theta(k)) - \mathcal{Q}_{t+1}^\pi(\bar{Z}_{t+1}^k)\right] + \mathcal{Q}_t^\pi(\bar{Z}_t) - F_t(\bar{Z}_t^k; \Theta(k)).$$

Secondly, we perform a change-of-feature as follows:

$$\left\langle \nabla F_t(\bar{Z}_t^k; \Theta(k)), \Theta(k) - \bar{\Theta}\right\rangle = \left\langle \nabla F_t(\bar{Z}_t^k; \Theta(0)), \Theta(k) - \bar{\Theta}\right\rangle + \mathrm{err}_{t,k}^{(1)}, \quad (34)$$

where

$$\mathrm{err}_{t,k}^{(1)} := \left\langle \nabla F_t(\bar{Z}_t^k; \Theta(k)) - \nabla F_t(\bar{Z}_t^k; \Theta(0)), \Theta(k) - \bar{\Theta}\right\rangle, \text{ and } |\mathrm{err}_{t,k}^{(1)}| \le \frac{2\beta_t^2\|\rho\|_2^2}{\sqrt{m}} \le \frac{2\beta_T^2\|\rho\|_2^2}{\sqrt{m}},$$

by Lemma B.1. Furthermore,

$$\left\langle \nabla F_t(\bar{Z}_t^k; \Theta(0)), \Theta(k) - \bar{\Theta}\right\rangle = F_t^{\mathsf{Lin}}(\bar{Z}_t^k; \Theta(k)) - F_t^{\mathsf{Lin}}(\bar{Z}_t^k; \bar{\Theta}), \quad (35)$$
$$= F_t(\bar{Z}_t^k; \Theta(k)) - \mathcal{Q}_t^\pi(\bar{Z}_t^k) + \mathrm{err}_{t,k}^{(2)} + \mathrm{err}_{t,k}^{(3)} \quad (36)$$

where

$$\mathrm{err}_{t,k}^{(2)} := F_t^{\mathsf{Lin}}(\bar{Z}_t^k; \Theta(k)) - F_t(\bar{Z}_t^k; \Theta(k)),$$
$$\mathrm{err}_{t,k}^{(3)} := -F_t^{\mathsf{Lin}}(\bar{Z}_t^k; \bar{\Theta}) + \mathcal{Q}_t^\pi(\bar{Z}_t^k).$$

Thus,

$$(\spadesuit)_t = -(\mathcal{Q}_t^\pi(\bar{Z}_t^k) - F_t(\bar{Z}_t^k; \Theta(k)))^2 + \gamma\check{\mathbb{E}}_t^k\left[F_{t+1}(\bar{Z}_{t+1}^k; \Theta(k)) - \mathcal{Q}_{t+1}^\pi(\bar{Z}_{t+1}^k)\right] \cdot (\mathcal{Q}_t^\pi(\bar{Z}_t^k) - F_t(\bar{Z}_t^k; \Theta(k)))$$
$$+ \check{\mathbb{E}}_t^k \delta_t(\bar{Z}_{t+1}^k; \Theta(k)) \sum_{j=1}^3 \mathrm{err}_{t,k}^{(j)}.$$

By equation 24, we have

$$\sup_{\bar{z} \in \bar{\mathbb{H}}_\infty} |\delta_t(\bar{z}_{t+1}; \Theta(k))| \le r_\infty + 2L_T\|\rho\|_2 =: \delta_{\mathsf{max}}$$

Now, let $\omega_{t,k} := \left(\mathbb{E}[(\mathcal{Q}_t^\pi(\bar{Z}_t^k) - F_t(\bar{Z}_t^k; \Theta(k)))^2]\right)^{1/2}$, where the expectation is over the joint distribution of $\Theta(k)$ and $\bar{Z}_T^k$. Then,

$$\mathbb{E}[(\spadesuit)_t] \le -\omega_{t,k}^2 + \gamma\omega_{t+1,k}\omega_{t,k} + \delta_{\mathsf{max}}\sum_{j=1}^3 \mathbb{E}|\mathrm{err}_{t,k}^{(j)}|.$$

From equation 25, we have

$$\mathbb{E}|\mathrm{err}_{t,k}^{(2)}| \le \frac{2}{\sqrt{m}}(\varrho_2\Lambda_T^2 + \varrho_1\chi_T)\|\rho\|_2^2.$$

From the approximation bound in Lemma B.2, we get

$$\mathbb{E}|\mathrm{err}_{t,k}^{(3)}| \le \sqrt{\mathbb{E}|\mathrm{err}_{t,k}^{(3)}|^2} \le \frac{2\|\nu\|_2\sqrt{1+\varrho_0^2} \cdot p_T(\alpha\varrho_1)}{\sqrt{m}}.$$

Also, note that $\omega_{t+1,k}\omega_{t,k} \le \frac{1}{2}(\omega_{t,k}^2 + \omega_{t+1,k}^2)$. Putting these together, we obtain the following bound for every $t \in \{0, 1, \dots, T-1\}$:

$$\mathbb{E}[(\spadesuit)_t] \le -\omega_{t,k}^2 + \frac{\gamma}{2}(\omega_{t+1,k}^2 + \omega_{t,k}^2) + \delta_{\mathsf{max}} \cdot \frac{C_T}{\sqrt{m}},$$

where

$$C_T := 2\beta_T^2 \|\rho\|_2^2 + 2(\varrho_2 \Lambda_T^2 + \varrho_1 \chi_T)\|\rho\|_2^2 + 2\|\nu\|_2\sqrt{1+\varrho_0^2} \cdot p_T(\alpha \varrho_1).$$

Hence, we obtain the following upper bound:

$$
\sum_{t=0}^{T-1} \gamma^t \mathbb{E}[(\spadesuit)_t] \le -(1-\gamma/2)\sum_{t<T}\gamma^t\omega_{t,k}^2 + \frac{\delta_{\mathsf{max}} \cdot C_T}{(1-\gamma)\sqrt{m}} + \underbrace{\frac{1}{2}\sum_{t<T}\gamma^{t+1}\omega_{t+1,k}^2}_{\le \frac{1}{2}\left(\sum_{t<T}\gamma^t\omega_{t,k}^2 + \gamma^T\omega_{T,k}^2\right)}
$$
$$
\le -\frac{1-\gamma}{2}\sum_{t<T}\gamma^t\omega_{t,k}^2 + \frac{1}{2}\gamma^T\omega_{T,k}^2 + \frac{C_T \cdot \delta_{\mathsf{max}}}{(1-\gamma)\sqrt{m}}. \tag{37}
$$

**Bounding** $\mathbb{E}[(\clubsuit)]$**.** Using the triangle inequality, we obtain:

$$\|\sum_{t<T}\gamma^t\delta_t(\bar{Z}_{t+1}^k;\Theta(k))\nabla F_t(\bar{Z}_t;\Theta(k))\|_2 \le \sum_{t<T}\gamma^t|\delta_t(\bar{Z}_{t+1}^k;\Theta(k))| \cdot \|\nabla F_t(\bar{Z}_t;\Theta(k))\|_2.$$

Since $\Theta(k) \in \Omega_{\rho,m}$ for every $k \in \mathbb{N}$ as a consequence of the max-norm regularization, we have

$$|\delta_t(\bar{Z}_{t+1}^k;\Theta(k))| \le \delta_{\mathsf{max}} = r_\infty + 2L_T\|\rho\|_2,$$
$$\|\nabla F_t(\bar{Z}_t^k;\Theta(k))\|_2^2 = \frac{1}{m}\sum_{i=1}^m \|\nabla_{\Theta_i}H_t^{(i)}(\bar{Z}_t^k;\Theta(k))\|_2^2 \le L_t^2 \le L_T^2,$$

for every $t < T$ with probability 1 since $\Theta_i \mapsto H_t^{(i)}(\bar{z}_t;\Theta_i)$ is $L_t$-Lipschitz continuous by Lemma B.1. Hence, we obtain:

$$\|\check{\nabla}\mathcal{R}_T(\bar{Z}_T^k;\Theta(k))\|_2 \le \frac{\delta_{\mathsf{max}}L_T}{1-\gamma}. \tag{38}$$

**Final step.** Now, taking expectation over $(\bar{Z}_t^k,\Theta(k))$ in equation 33, and substituting equation 37 and equation 38, we obtain:

$$\mathbb{E}[\Psi(\Theta(k+1)) - \Psi(\Theta(k))] \le -\eta(1-\gamma)\sum_{t=0}^{T-1}\gamma^t\omega_{t,k}^2 + \eta\gamma^T\omega_{T,k}^2 + \eta\frac{\delta_{\mathsf{max}} \cdot C_T}{(1-\gamma)\sqrt{m}} + \eta^2\frac{\delta_{\mathsf{max}}^2 L_T^2}{(1-\gamma)^2},$$

for every $k \in \mathbb{N}$. Note that $\Psi(\Theta(0)) \le \|\nu\|_2^2$. Thus, telescoping sum over $k = 0, 1, \dots, K-1$ yields

$$\frac{1}{K}\sum_{k=0}^{K-1}\mathcal{R}_T(\Theta(k)) \le \frac{\|\nu\|_2^2}{\eta(1-\gamma)K} + \frac{\eta\delta_{\mathsf{max}}^2 L_T^2}{(1-\gamma)^3} + \frac{\delta_{\mathsf{max}} \cdot C_T}{(1-\gamma)^2\sqrt{m}} + \frac{\gamma^T}{(1-\gamma)K}\sum_{k=0}^{K-1}\omega_{T,k}^2. \tag{39}$$

The final inequality in the proof stems from the linearization result Lemma B.2, and directly follows from

$$\mathcal{R}_T\left(\frac{1}{K}\sum_{k<K}\Theta(k)\right) \le \frac{4}{K}\sum_{k<K}\mathcal{R}_T(\Theta(k)) + \frac{6}{\sqrt{m}}\left(\varrho_2\Lambda_T^2 + \varrho_1\chi_T\right)\|\rho\|_2^2,$$

which directly follows from Cayci & Eryilmaz (2025), Corollary 1. $\qquad\square$

In the following, we study the error under mean-path Rec-TD learning algorithm.

**Theorem B.4** (Finite-time bounds for mean-path Rec-TD). *For $K \in \mathbb{N}$, with the step-size choice $\eta = \frac{(1-\gamma)^2}{64L_T^2}$, mean-path Rec-TD learning achieves the following error bound:*

$$\mathbb{E}\left[\frac{1}{K}\sum_{k<K}\mathcal{R}_T^\pi(\Theta(k))\right] \leq \frac{2\|\nu\|_2^2}{(1-\gamma)\eta K} + \frac{\gamma^T \omega_{T,k}}{1-\gamma} + \frac{C_T \delta_{\mathsf{max}}}{(1-\gamma)^2\sqrt{m}} + \eta\left(\frac{(C_T')^2}{m} + 16\gamma^{2T}L_T^4(\|\rho\|_2^2 + \|\nu\|_2^2)\right),$$

*where $C_T'$ and $L_T$ are terms that do not depend on $K$.*

Theorem B.4 indicates that if a noiseless semi-gradient is used in Rec-TD, then the rate can be improved from $\mathcal{O}\left(\frac{1}{\sqrt{K}}\right)$ to $\mathcal{O}\left(\frac{1}{K}\right)$, indicating the potential limits of using variance-reduction schemes.

*Proof of Theorem B.4.* At any iteration $k \in \mathbb{N}$, let

$$\bar{\nabla}\mathcal{R}_T(\Theta(k)) := \mathbb{E}_\mu^\pi\left[\check{\nabla}\mathcal{R}(\bar{Z}_t^k; \Theta(k))\right], \tag{40}$$

be the **mean-path semi-gradient**. First, note that

$$\|\bar{\nabla}\mathcal{R}_T(\Theta(k))\|_2^2 \leq 2\|\bar{\nabla}\mathcal{R}_T(\Theta(k)) - \bar{\nabla}\mathcal{R}_T(\bar{\Theta})\|_2^2 + 2\|\bar{\nabla}\mathcal{R}_T(\bar{\Theta})\|_2^2. \tag{41}$$

**Bounding $\|\bar{\nabla}\mathcal{R}_T(\bar{\Theta})\|_2^2$.** For any $k \in \mathbb{N}, t \leq T$, we have

$$\mathbb{E}\left[\delta_t(\bar{Z}_{t+1}^k; \bar{\Theta})|\bar{Z}_t^k, \Theta(0), c\right] = \gamma\mathbb{E}[F_{t+1}(\bar{Z}_{t+1}^k; \bar{\Theta}) - \mathcal{Q}_{t+1}^\pi(\bar{Z}_{t+1}^k)|\bar{Z}_t^k, \Theta(0), c] + \mathcal{Q}_t^\pi(\bar{Z}_t^k) - F_t(\bar{Z}_t^k; \bar{\Theta}).$$

Since $\|\nabla F_t(\bar{z}_t; \bar{\Theta})\|_2 \leq L_t$, the following inequality holds:

$$\begin{aligned}
\left\|\mathbb{E}\left[\delta_t(\bar{Z}_{t+1}^k; \bar{\Theta})\nabla F_t(\bar{Z}_t^k; \bar{\Theta})\right]\right\|_2 &\leq \mathbb{E}\left\|\mathbb{E}\left[\delta_t(\bar{Z}_{t+1}^k; \bar{\Theta})|\bar{Z}_t^k, \Theta(0), c\right]\nabla F_t(\bar{Z}_t^k; \bar{\Theta})\right\|_2, \\
&\leq L_T\mathbb{E}\left|\mathbb{E}\left[\delta_t(\bar{Z}_{t+1}^k; \bar{\Theta})|\bar{Z}_t^k, \Theta(0), c\right]\right|, \\
&\leq L_T\left(\gamma\mathbb{E}\left|F_{t+1}(\bar{Z}_{t+1}^k; \bar{\Theta}) - \mathcal{Q}_{t+1}^\pi(\bar{Z}_{t+1}^k)\right| + \mathbb{E}\left|\mathcal{Q}_t^\pi(\bar{Z}_t^k) - F_t(\bar{Z}_t^k; \bar{\Theta})\right|\right), \tag{42}
\end{aligned}$$

where we used Jensen's inequality, the law of iterated expectations, and triangle inequality. From the above inequality, we obtain

$$\begin{aligned}
\|\bar{\nabla}\mathcal{R}_T(\bar{\Theta})\|_2 &\overset{\text{①}}{\leq} \sum_{t=0}^{T-1}\gamma^t\left\|\mathbb{E}\left[\delta_t(\bar{Z}_{t+1}^k; \bar{\Theta})\nabla F_t(\bar{Z}_t^k; \bar{\Theta})\right]\right\|_2, \\
&\overset{\text{②}}{\leq} L_T\gamma\sum_{t<T}\gamma^t\mathbb{E}|F_{t+1}(\bar{Z}_{t+1}^k; \bar{\Theta}) - \mathcal{Q}_{t+1}^\pi(\bar{Z}_{t+1}^k)| + L_T\sum_{t<T}\gamma^t\mathbb{E}|\mathcal{Q}_t^\pi(\bar{Z}_t^k) - F_t(\bar{Z}_t^k; \bar{\Theta})|, \\
&\overset{\text{③}}{\leq} \frac{L_T}{\sqrt{1-\gamma}}\left(\gamma\mathbb{E}\sqrt{\sum_{t<T}\gamma^t|F_{t+1}(\bar{Z}_{t+1}^k; \bar{\Theta}) - \mathcal{Q}_{t+1}^\pi(\bar{Z}_{t+1}^k)|^2} + \mathbb{E}\sqrt{\sum_{t<T}\gamma^t|F_t(\bar{Z}_t^k; \bar{\Theta}) - \mathcal{Q}_t^\pi(\bar{Z}_t^k)|^2}\right), \\
&\overset{\text{④}}{\leq} \frac{L_T}{\sqrt{1-\gamma}}\left(\gamma\sqrt{\mathbb{E}\sum_{t<T}\gamma^t|F_{t+1}(\bar{Z}_{t+1}^k; \bar{\Theta}) - \mathcal{Q}_{t+1}^\pi(\bar{Z}_{t+1}^k)|^2} + \sqrt{\mathbb{E}\sum_{t<T}\gamma^t|F_t(\bar{Z}_t^k; \bar{\Theta}) - \mathcal{Q}_t^\pi(\bar{Z}_t^k)|^2}\right), \\
&\overset{\text{⑤}}{\leq} \frac{\sqrt{2}(1+\gamma)L_T}{\sqrt{1-\gamma}}\frac{\|\nu\|_2\sqrt{1+\varrho_0^2}\cdot p_T(\varrho_1\alpha)}{\sqrt{m}}.
\end{aligned}$$

where ① follows from triangle inequality, ② follows from equation 42, ③ follows from Cauchy-Schwarz inequality and the monotonicity of the geometric series $T \mapsto \sum_{t<T}\gamma^t$, ④ follows from Jensen's inequality, and finally ⑤ follows from Lemma B.2. Hence, we obtain

$$\|\bar{\nabla}\mathcal{R}_T(\bar{\Theta})\|_2^2 \leq \frac{8L_T^2\|\nu\|_2^2(1+\varrho_0^2)p_T^2(\varrho_1\alpha)}{(1-\gamma)m}. \tag{43}$$

**Bounding** $\|\bar{\nabla}\mathcal{R}_T(\Theta(k)) - \bar{\nabla}\mathcal{R}_T(\bar{\Theta})\|_2^2$**.** First, note that

$$\|\bar{\nabla}\mathcal{R}_T(\Theta(k)) - \bar{\nabla}\mathcal{R}_T(\bar{\Theta})\|_2 = \|\mathbb{E}\Big[\sum_{t<T}\gamma^t\big(\delta_t(\bar{Z}_{t+1}^k;\Theta(k))\nabla F_t(\bar{Z}_t^k;\Theta(k)) - \delta_t(\bar{Z}_{t+1}^k;\bar{\Theta})\nabla F_t(\bar{Z}_t^k;\bar{\Theta})\big)\Big]\|\|_2$$

We make the following decomposition for each $t < T$:

$$\delta_t(\bar{Z}_{t+1}^k;\Theta(k))\nabla F_t(\bar{Z}_t^k;\Theta(k)) - \delta_t(\bar{Z}_{t+1}^k;\bar{\Theta})\nabla F_t(\bar{Z}_t^k;\bar{\Theta}) = \delta_t(\bar{Z}_{t+1}^k;\Theta(k))\big(\nabla F_t(\bar{Z}_t^k;\Theta(k)) - \nabla F_t(\bar{Z}_t^k;\bar{\Theta})\big)$$
$$+ \nabla F_t(\bar{Z}_t^k;\Theta(k))\big(\delta_t(\bar{Z}_{t+1}^k;\bar{\Theta}) - \delta_t(\bar{Z}_{t+1}^k;\Theta(k))\big) \quad (44)$$

By Lemma B.1, we have $|\delta_t(\bar{Z}_{t+1}^k;\Theta)| \leq \delta_{\mathsf{max}}$ and $\|\nabla F_t(\bar{Z}_t^k;\Theta)\|_1 \leq L_t \leq L_T$ almost surely for any $\Theta \in \Omega_{\rho,m}$, which holds for $\Theta(k)$ (due to the max-norm projection) and $\bar{\Theta}$. As such, by triangle inequality,

$$\|\bar{\nabla}\mathcal{R}_T(\Theta(k)) - \bar{\nabla}\mathcal{R}_T(\bar{\Theta})\|_2 \leq \sum_{t<T}\gamma^t\Big(\delta_{\mathsf{max}}\frac{\beta_t^2\mathbb{E}\|\Theta(k)-\bar{\Theta}\|_2^2}{m} + L_t\mathbb{E}|\delta_t(\bar{Z}_{t+1}^k;\bar{\Theta}) - \delta_t(\bar{Z}_{t+1}^k;\Theta(k))|\Big),$$

$$\leq \underbrace{\frac{\delta_{\mathsf{max}}\beta_T^2(\|\rho\|_2^2 + \|\nu\|_2^2)}{m(1-\gamma)}}_{=:\frac{C_T^{(4)}}{m}} + L_T\mathbb{E}\Big[\sum_{t=0}^{T-1}\gamma^t|\delta_t(\bar{Z}_{t+1}^k;\bar{\Theta}) - \delta_t(\bar{Z}_{t+1}^k;\Theta(k))|\Big] \quad (45)$$

Note that

$$\sum_{t<T}\gamma^t|\delta_t(\bar{Z}_{t+1}^k;\Theta(k)) - \delta_t(\bar{Z}_{t+1}^k;\bar{\Theta})| = \sum_{t<T}\gamma^t\Big(|F_{t+1}(\bar{Z}_{t+1}^k;\bar{\Theta}) - F_{t+1}(\bar{Z}_{t+1}^k;\Theta(k))| + |F_t(\bar{Z}_t^k;\bar{\Theta}) - F_t(\bar{Z}_t^k;\Theta(k))|\Big),$$

$$\leq 2\sum_{t<T}\gamma^t\Big|F_t(\bar{Z}_t^k;\bar{\Theta}) - F_t(\bar{Z}_t^k;\Theta(k))\Big| + \gamma^T L_T\|\Theta(k) - \bar{\Theta}\|_2, \quad (46)$$

where the second line follows from the Lipschitz continuity of $\Theta \mapsto F_t(\cdot;\Theta)$. Then, adding and subtracting $\mathcal{Q}_t^\pi$ to each term, we obtain

$$\sum_{t<T}\gamma^t|\delta_t(\bar{Z}_{t+1}^k;\Theta(k)) - \delta_t(\bar{Z}_{t+1}^k;\bar{\Theta})| \leq 2\sum_{t<T}\gamma^t\big(|F_t(\bar{Z}_t^k;\bar{\Theta}) - \mathcal{Q}_t^\pi(\bar{Z}_t^k)| + |\mathcal{Q}_t^\pi(\bar{Z}_t^k) - F_t(\bar{Z}_t^k;\Theta(k))|\big)$$
$$+ \gamma^T L_T\|\Theta(k) - \bar{\Theta}\|_2. \quad (47)$$

Taking expectation, we obtain

$$\mathbb{E}\sum_{t<T}\gamma^t|\delta_t(\bar{Z}_{t+1}^k;\Theta(k)) - \delta_t(\bar{Z}_{t+1}^k;\bar{\Theta})| \leq \frac{2}{\sqrt{1-\gamma}}\sqrt{\mathbb{E}\Big[\sum_{t<T}\gamma^t|F_t(\bar{Z}_t^k;\Theta(k)) - \mathcal{Q}_t^\pi(\bar{Z}_t^k)|^2\Big]}$$

$$+ \frac{2}{\sqrt{1-\gamma}}\sqrt{\mathbb{E}\Big[\sum_{t<T}\gamma^t|F_t(\bar{Z}_t^k;\bar{\Theta}) - \mathcal{Q}_t^\pi(\bar{Z}_t^k)|^2\Big]} + \gamma^T L_T\|\Theta(k) - \bar{\Theta}\|_2.$$

By Lemma B.2 and equation 25, we have

$$\mathbb{E}|F_t(\bar{Z}_t^k;\bar{\Theta}) - \mathcal{Q}_t^\pi(\bar{Z}_t^k)|^2 \leq \frac{4}{m}\|\nu\|_2^2\varrho_1^2(1+\varrho_0)^2 p_t^2(\alpha\varrho_1) + \frac{4}{m}(\varrho_2\Lambda_T^2 + \varrho_1\chi_T)^2\|\rho\|_2^4,$$

for any $t < T$. Thus,

$$\mathbb{E}\sum_{t<T}\gamma^t|\delta_t(\bar{Z}_{t+1}^k;\Theta(k)) - \delta_t(\bar{Z}_{t+1}^k;\bar{\Theta})| \leq \frac{2}{\sqrt{1-\gamma}}\sqrt{\mathbb{E}\Big[\sum_{t<T}\gamma^t|F_t(\bar{Z}_t^k;\Theta(k)) - \mathcal{Q}_t^\pi(\bar{Z}_t^k)|^2\Big]}$$

$$+ \frac{1}{\sqrt{m}}\underbrace{\frac{4}{\sqrt{(1-\gamma)^3}}\big(\|\nu\|_2\varrho_1(1+\varrho_0)p_T(\alpha\varrho_1) + (\varrho_2\Lambda_T^2 + \varrho_1\chi_T)\|\rho\|_2^2\big)}_{=:C_T^{(3)}} + \gamma^T L_T\underbrace{\|\Theta(k) - \bar{\Theta}\|_2}_{\leq \|\rho\|_2 + \|\nu\|_2}.$$

This results in the following bound:

$$\mathbb{E}\sum_{t<T}\left[\gamma^t|\delta_t(\bar{Z}_{t+1}^k;\Theta(k)) - \delta_t(\bar{Z}_{t+1}^k;\bar{\Theta})|\right] \le \frac{2}{\sqrt{1-\gamma}}\sqrt{\mathcal{R}_T(\Theta(k))} + \frac{C_T^{(3)}}{\sqrt{m}} + \gamma^T L_T(\|\rho\|_2 + \|\nu\|_2). \qquad (48)$$

Substituting the local smoothness result in equation 48 into equation 45, we obtain

$$\|\bar{\nabla}\mathcal{R}_T(\Theta(k)) - \bar{\nabla}\mathcal{R}_T(\bar{\Theta})\|_2 \le L_T\left(\frac{2}{\sqrt{1-\gamma}}\sqrt{\mathcal{R}_T(\Theta(k))} + \frac{C_T^{(3)}}{\sqrt{m}} + \gamma^T L_T(\|\rho\|_2 + \|\nu\|_2)\right) + \frac{C_T^{(4)}}{m}.$$

Thus, we obtain

$$\|\bar{\nabla}\mathcal{R}_T(\Theta(k)) - \bar{\nabla}\mathcal{R}_T(\bar{\Theta})\|_2^2 \le \frac{16L_T^2}{1-\gamma}\mathcal{R}_T(\Theta(k)) + \frac{4(C_T^{(3)})^2 L_T^2 + 4(C_T^{(4)})^2}{m} + 8\gamma^{2T}L_T^4(\|\rho\|_2^2 + \|\nu\|_2^2). \qquad (49)$$

Using equation 43 and equation 49 together, we obtain

$$\|\bar{\nabla}\mathcal{R}_T(\Theta(k))\|_2^2 \le 2\|\bar{\nabla}\mathcal{R}_T(\Theta(k)) - \bar{\nabla}\mathcal{R}_T(\bar{\Theta})\|_2^2 + 2\|\bar{\nabla}\mathcal{R}_T(\bar{\Theta})\|_2^2,$$
$$\le \frac{32L_T^2\mathcal{R}_T(\Theta(k))}{1-\gamma} + \frac{(C_T')^2}{m} + 16\gamma^{2T}L_T^4(\|\rho\|_2^2 + \|\nu\|_2^2). \qquad (50)$$

In the final step, we use equation 33, equation 37 and equation 50 together:

$$\mathbb{E}\left[\Psi(\Theta(k+1)) - \Psi(\Theta(k))\right] \le -\eta(1-\gamma)\mathbb{E}\mathcal{R}_T(\Theta(k)) + \eta\gamma^T\omega_{T,k} + \eta\frac{C_T\delta_{\mathsf{max}}}{(1-\gamma)\sqrt{m}}$$
$$+ \eta^2\left(\frac{32L_T^2\mathbb{E}\mathcal{R}_T(\Theta(k))}{1-\gamma} + \frac{(C_T')^2}{m} + 16\gamma^{2T}L_T^4(\|\rho\|_2^2 + \|\nu\|_2^2)\right), \qquad (51)$$

where the expectation is over the random initialization. Choosing $\eta = \frac{(1-\gamma)^2}{64L_T^2}$, we obtain

$$\mathbb{E}[\Psi(\Theta(k+1)) - \Psi(\Theta(k))] \le -\frac{\eta(1-\gamma)}{2}\mathbb{E}\mathcal{R}_T(\Theta(k)) + \eta\gamma^T\omega_{T,k} + \eta\frac{C_T\delta_{\mathsf{max}}}{(1-\gamma)\sqrt{m}}$$
$$+ \eta^2\left(\frac{(C_T')^2}{m} + 16\gamma^{2T}L_T^4(\|\rho\|_2^2 + \|\nu\|_2^2)\right). \qquad (52)$$

Telescoping sum over $k = 0, 1, \ldots, K-1$, and re-arranging terms, we obtain:

$$\mathbb{E}\left[\frac{1}{K}\sum_{k<K}\mathcal{R}_T(\Theta(k))\right] \le \frac{2\|\nu\|_2^2}{(1-\gamma)\eta K} + \frac{\gamma^T\omega_{T,k}}{1-\gamma} + \frac{C_T\delta_{\mathsf{max}}}{(1-\gamma)^2\sqrt{m}} + \eta\left(\frac{(C_T')^2}{m} + 16\gamma^{2T}L_T^4(\|\rho\|_2^2 + \|\nu\|_2^2)\right). \qquad (53)$$

$\square$

## C  Numerical Experiments for Rec-TD

In the following, we will demonstrate the numerical performance of Rec-TD for a given non-stationary policy $\pi^{\mathsf{greedy}}$.

**POMDP setting.** We consider a randomly-generated finite POMDP instance with $|\mathbb{S}| = |\mathbb{Y}| = 8$, $|\mathbb{A}| = 4$, $r(s,a) \sim \mathsf{Unif}[0,1]$ for all $(s,a) \in \mathbb{S} \times \mathbb{A}$. For a fixed ambient dimension $d = 8$, we use a random feature mapping $(y,a) \mapsto \varphi(y,a) \sim \mathcal{N}(0, I_d)$, $\forall(y,a) \sim \mathbb{Y} \times \mathbb{A}$.

**$\epsilon$-greedy policy.** Let

$$j^\star(t) \in \arg\max_{0 \le j < t} r_j,$$

be the instance before $t$ at which the maximum reward was obtained, and let

$$\pi_t^{\epsilon-\text{greedy}}(a|Z_t) = \begin{cases} \frac{1}{|\mathbb{A}|}, & \text{w.p. } \min\{\frac{2+t}{10}, p_{\text{exp}}\}, \\ \mathbb{1}_{a=A_{j^\star(t)}}, & \text{w.p. } 1 - \min\{\frac{2+t}{10}, p_{\text{exp}}\}, \end{cases} \tag{54}$$

be the greedy policy with a user-specified exploration probability $p_{\text{exp}} \in (0,1)$. The long-term dependencies in this greedy policy are obviously controlled by $p_{\text{exp}}$: a small exploration probability will make the policy (thus, the corresponding $\mathcal{Q}$-functions) more history-dependent. Since the exact computation of $(\mathcal{Q}_t^\pi)_{t\in\mathbb{N}}$ is highly intractable for POMDPs, we use (empirical) mean-squared temporal difference (MSTD) [2] as a surrogate loss.

**Example 1 (Short-term memory).** We first consider the performance of Rec-TD with learning rate $\eta = 0.05$, discount factor $\gamma = 0.9$ and RNNs with various choices of network width $m$. For $p_{\text{exp}} = 0.8$, the performance of Rec-TD is demonstrated in Figure 2. Consistent with the theoretical results in Theorem

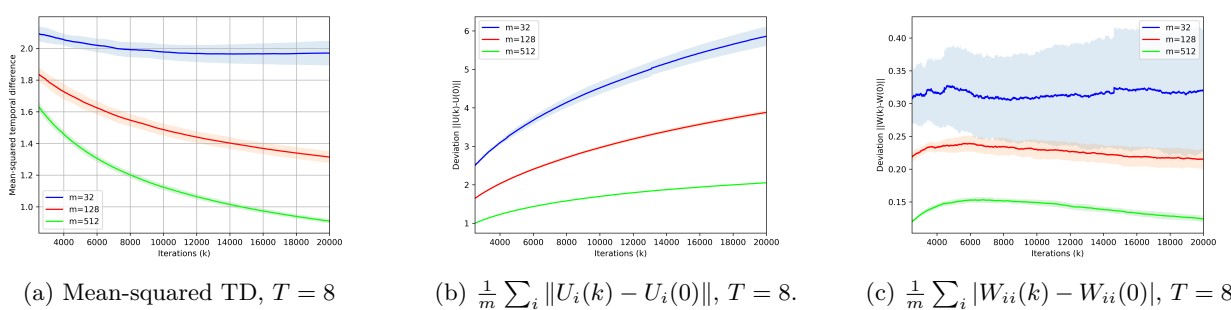

(a) Mean-squared TD, $T = 8$     (b) $\frac{1}{m}\sum_i \|U_i(k) - U_i(0)\|$, $T = 8$.     (c) $\frac{1}{m}\sum_i |W_{ii}(k) - W_{ii}(0)|$, $T = 8$.

Figure 2: Mean-squared TD and (mean) parameter deviation under Rec-TD for the case $p_{\text{exp}} = 0.8$ and $\gamma = 0.9$. The mean curve and confidence intervals (90%) stem from 5 trials.

5.4, Rec-TD (1) achieves smaller error with larger network width $m$, (2) requires smaller deviation from the random initialization $\Theta(0)$, which is known as the *lazy training* phenomenon.

**Example 2 (Long-term memory).** In the second example, we consider the same POMDP with same random samples, and an RNN with the same neural network initialization. The exploration probability is reduced to $p_{\text{exp}} = 0.25$, which leads to longer dependency on the history. This impact can be observed in Figure 3c, which implies a larger spectral radius compared to Example 1 (in comparison with Figure 2c).

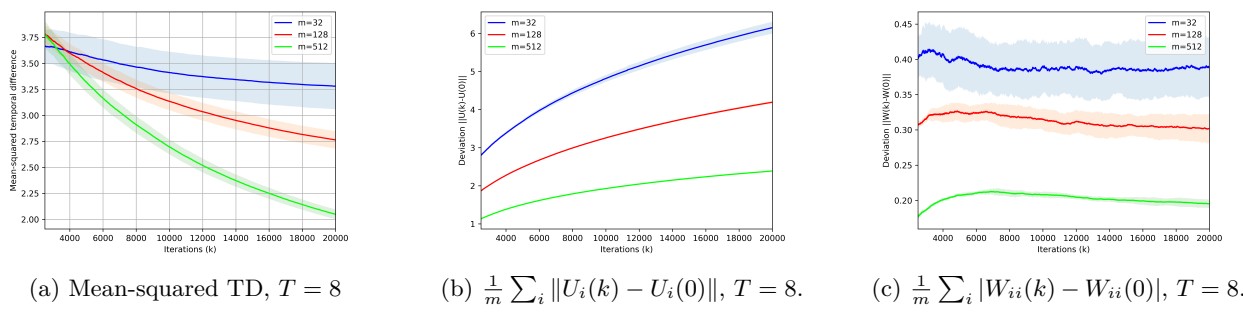

(a) Mean-squared TD, $T = 8$     (b) $\frac{1}{m}\sum_i \|U_i(k) - U_i(0)\|$, $T = 8$.     (c) $\frac{1}{m}\sum_i |W_{ii}(k) - W_{ii}(0)|$, $T = 8$.

Figure 3: Mean-squared TD and (mean) parameter deviation under Rec-TD for the case $p_{\text{exp}} = 0.25$ and $\gamma = 0.9$. The mean curve and confidence intervals (90%) stem from 5 trials.

In Figure 4, we investigate the impact of the truncation level $T$ on the MSTD performance with $p_{\text{exp}} = 0.25$, which implies long-term dependency, for an RNN with $m = 256$ units. Increasing $T$ implies a larger MSTD due to long-term dependencies, validating the theoretical results.

---

[2]the empirical mean of independently sampled $\left\{ \frac{1}{k}\sum_{s<k} \hat{\mathcal{R}}_T^{\text{TD}}(\Theta(s)) : k \in \mathbb{N} \right\}$ where $\hat{\mathcal{R}}_T^{\text{TD}}(\Theta(k)) = \sum_{t=0}^{T-1} \gamma^t \delta_t^2(\bar{Z}_t^k; \Theta(k))$.

## D  Policy Gradients under Partial Observability

In this section, we will provide basic results for policy gradients under POMDPs, which is critical to develop the natural policy gradient method for POMDPs.

**Proposition D.1.** *Let $\pi' \in \Pi_{\mathsf{NM}}$ be an admissible policy, and let $\bar{Z}_T \sim P_T^{\pi',\mu}$. Then, for any $t < T$, conditional distribution of $S_t$ given $\bar{Z}_t$ is independent of $\pi'$. Furthermore, for any $\pi \in \Pi_{\mathsf{NM}}$, the conditional distribution of $r(S_t, A_t) + \gamma \mathcal{V}_{t+1}^\pi(Z_{t+1})$ given $\bar{Z}_t$ is independent of $\pi'$.*

*Proof of Prop. D.1.* Let the belief at time $t \in \mathbb{N}$ be defined as

$$b_t(s) := \mathbb{P}(S_t = s | \bar{Z}_t). \tag{55}$$

For any non-stationary admissible policy $\pi$, the belief function is policy-independent. To see this, note that

$$\mathbb{P}(S_t = s_t, \bar{Z}_t = \bar{z}_t) = \sum_{(s_0,\ldots,s_{t-1})\in\mathbb{S}^t} \mathbb{P}(S_0 = s_0 | Y_0 = y)\pi_0(a_0|z_0)\prod_{k=0}^{t-1}\mathcal{P}(s_{k+1}|s_k, a_k)\phi(y_{k+1}|s_{k+1})\pi_{k+1}(a_{k+1}|z_{k+1}),$$

$$= \left(\prod_{k=0}^{t}\pi_k(a_k|z_k)\right)\sum_{(s_0,\ldots,s_{t-1})\in\mathbb{S}^t}\mathbb{P}(S_0 = s_0 | Y_0 = y)\prod_{k=0}^{t-1}\mathcal{P}(s_{k+1}|s_k, a_k)\phi(y_{k+1}|s_{k+1}),$$

since $\prod_{k=0}^{t}\pi_k(a_k|z_k)$ does not depend on the summands $(s_0,\ldots,s_{t-1})$ – note that we use the notation $\mathcal{P}(s_{k+1}|s_k, a_k) := \mathcal{P}(s_k, a_k, \{S_{k+1} = s_{k+1}\})$ and $\phi(y_k|s_k) := \phi(s_k, \{Y_k = y_k\})$. Thus,

$$b_t(s_t) = \frac{\sum_{(s_0,\ldots,s_{t-1})\in\mathbb{S}^t}\mathbb{P}(S_0 = s_0 | Y_0 = y)\prod_{k=0}^{t-1}\mathcal{P}(s_{k+1}|s_k, a_k)\phi(y_{k+1}|s_{k+1})}{\sum_{(s'_0,\ldots,s'_{t-1},s'_t)\in\mathbb{S}^{t+1}}\mathbb{P}(S_0 = s'_0 | Y_0 = y)\prod_{k=0}^{t-1}\mathcal{P}(s'_{k+1}|s'_k, a_k)\phi(y_{k+1}|s'_{k+1})},$$

independent of $\pi$. As such, we have

$$\mathbb{E}^{\pi'}[r_t + \gamma\mathcal{V}^\pi(Z_{t+1})|\bar{Z}_t] = \sum_{s\in\mathbb{S}}b_t(s)\mathbb{E}^{\pi'}[r_t + \gamma\mathcal{V}_{t+1}^\pi(Z_{t+1})|\bar{Z}_t = \bar{z}_t, S_t = s],$$

$$= \sum_{s_t,s_{t+1}\in\mathbb{S}}\sum_{y\in\mathbb{Y}}b_t(s_t)\left(r(s_t, A_t) + \gamma\mathcal{P}(s_{t+1}|s_t, A_t)\phi(y|s_{t+1})\mathcal{V}_{t+1}^\pi(Z_t, y_{t+1})\right),$$

$$= \mathbb{E}[r_t + \gamma\mathcal{V}_{t+1}^\pi(Z_{t+1})|\bar{Z}_t = \bar{z}_t],$$

in other words, the conditional distribution of $r(S_t, A_t) + \gamma\mathcal{V}_{t+1}^\pi(Z_{t+1})$ given $\{\bar{Z}_t = \bar{z}_t\}$ is independent of $\pi'$. We also know from Prop. B.3 that

$$\mathbb{E}^{\pi'}[r_t + \gamma\mathcal{V}_{t+1}^\pi(Z_{t+1})|\bar{Z}_t = \bar{z}_t] = \mathbb{E}[r_t + \gamma\mathcal{V}_{t+1}^\pi(Z_{t+1})|\bar{Z}_t = \bar{z}_t] = \mathcal{Q}_t^\pi(\bar{z}_t).$$

$\square$

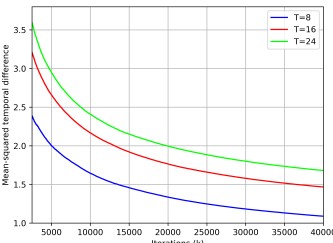

Figure 4: MSTD performance with $m = 256$ with various sequence lengths $T$ with $p_{\mathsf{exp}} = 0.25$. Increasing $T$ implies larger MSTD.

The next result generalizes the policy gradient theorem to POMDPs. We note that there is an extension of REINFORCE-type policy gradient for POMDPs in Wierstra et al. (2010). The following result is a different and improved version as it ① provides a variance-reduced unbiased estimate of the policy gradient for POMDPs, and more importantly ② yields the compatible function approximation (Prop. 6.1) that yields natural policy gradient (NPG) for POMDPs.

**Proposition D.2** (Policy gradient – POMDPs). *For any $\Phi \in \mathbb{R}^{m(d+1)}$, we have*

$$\nabla_\Phi \mathcal{V}^{\pi^\Phi}(\mu) = \mathbb{E}_\mu^{\pi^\Phi}\left[\sum_{t=0}^\infty \gamma^t \cdot \mathcal{Q}_t^{\pi^\Phi}(Z_t, A_t) \cdot \nabla_\Phi \ln \pi_t^\Phi(A_t|Z_t)\right], \tag{56}$$

*for any $\mu \in \Delta(\mathbb{Y})$.*

*Proof of Prop. D.2.* For any $t \in \mathbb{N}$, we have

$$\mathcal{V}_t^{\pi^\Phi}(z_t) = \sum_{a_t} \pi_t^\Phi(a_t|z_t) \mathcal{Q}_t^{\pi^\Phi}(z_t, a_t), \tag{57}$$

by Prop. B.3. Thus, we obtain

$$\nabla \mathcal{V}_t^{\pi^\Phi}(z_t) = \sum_{a_t} \pi_t^\Phi(a_t|z_t) \nabla \ln \pi_t^\Phi(a_t|z_t) \mathcal{Q}_t^{\pi^\Phi}(z_t, a_t) + \sum_{a_t} \pi_t^\Phi(a_t|z_t) \nabla \mathcal{Q}_t^{\pi^\Phi}(z_t, a_t),$$

$$= \mathbb{E}^{\pi^\Phi}[\nabla \ln \pi_t^\Phi(A_t|Z_t) \mathcal{Q}_t^{\pi^\Phi}(Z_t, A_t) + \nabla \mathcal{Q}_t^{\pi^\Phi}(Z_t, A_t)|Z_t = z_t]. \tag{58}$$

Now, note that

$$\mathcal{Q}_t^{\pi^\Phi}(z_t, a_t) = \mathbb{E}[r(S_t, A_t) + \gamma \mathcal{V}_{t+1}^{\pi^\Phi}(Z_{t+1})|\bar{Z}_t = (z_t, a_t)],$$

$$= \sum_{s_t} b_t(s_t) \left(r(s_t, a_t) + \gamma \sum_{s_{t+1}} \mathcal{P}(s_{t+1}|s_t, a_t) \sum_{y_{t+1}} \phi(y_{t+1}|s_{t+1}) \mathcal{V}_{t+1}^{\pi^\Phi}(z_{t+1})\right),$$

where $z_{t+1} = (z_t, a_t, y_{t+1})$. As a consequence of Prop. D.1, we have $\nabla_\Phi \sum_{s_t} b_t(s_t) r(s_t, a_t) = 0$, and also

$$\nabla_\Phi \mathcal{Q}_t^{\pi^\Phi}(z_t, a_t) = \gamma \sum_{s_t} b_t(s_t) \sum_{s_{t+1}} \mathcal{P}(s_{t+1}|s_t, a_t) \sum_{y_{t+1}} \phi(y_{t+1}|s_{t+1}) \nabla_\Phi \mathcal{V}_{t+1}^{\pi^\Phi}(z_{t+1}),$$

$$= \gamma \mathbb{E}[\nabla \ln \pi_{t+1}^\Phi(A_{t+1}|Z_{t+1}) \mathcal{Q}_{t+1}^{\pi^\Phi}(Z_{t+1}, A_{t+1}) + \nabla_\Phi \mathcal{Q}_{t+1}^{\pi^\Phi}(Z_{t+1}, A_{t+1})|\bar{Z}_t = (z_t, a_t)],$$

$$= \gamma \mathbb{E}^{\pi^\Phi}\left[\sum_{k=t+1}^\infty \gamma^{k-t-1} \nabla_\Phi \ln \pi_k^\Phi(A_k|Z_k) \mathcal{Q}_k^{\pi^\Phi}(Z_k, A_k)\Big|\bar{Z}_t = (z_t, a_t)\right].$$

Using the above recursive formula for $\nabla_\Phi \mathcal{Q}_t^{\pi^\Phi}$ along with the law of iterated expectations in equation 58, we obtain

$$\nabla_\Phi \mathcal{V}_t^{\pi^\Phi}(z_t) = \mathbb{E}^{\pi^\Phi}\left[\sum_{k=t}^\infty \gamma^{k-t} \nabla_\Phi \ln \pi_k^\Phi(A_k|Z_k) \mathcal{Q}_k^{\pi^\Phi}(Z_k, A_k)\Big|Z_t = z_t\right]. \tag{59}$$

Since we have $\mathcal{V}^\pi := \mathcal{V}_0^\pi$, and also $\nabla_\Phi \mathcal{V}^{\pi^\Phi}(\mu) = \nabla_\Phi \sum_{z_0} \mu(z_0) \mathcal{V}^{\pi^\Phi}(z_0) = \sum_{z_0} \mu(z_0) \nabla_\Phi \mathcal{V}^{\pi^\Phi}(z_0)$ by the linearity of gradient, we conclude the proof.

**Note on the baseline.** Similar to the case of fully-observable MDPs, adding a baseline $q_t^{\pi^\Phi}(z_t)$ to the $\mathcal{Q}$-function does not change the policy gradients since $\sum_a \pi_t(a|z_t) \nabla \ln \pi_t^\Phi(a|z_t) q_t^{\pi^\Phi}(z_t) = q_t^{\pi^\Phi}(z_t) \sum_a \nabla \pi_t^\Phi(a|z_t) = q_t^{\pi^\Phi}(z_t) \nabla \sum_a \pi_t^\Phi(a|z_t) = 0$. Thus, we also have

$$\nabla_\Phi \mathcal{V}^{\pi^\Phi}(\mu) = \mathbb{E}_\mu^{\pi^\Phi}\left[\sum_{t=0}^\infty \gamma^t \mathcal{A}_t^{\pi^\Phi}(Z_t, A_t) \nabla_\Phi \ln \pi_t^\Phi(A_t|Z_t)\right], \tag{60}$$

which uses $q_t^{\pi^\Phi} = \mathcal{V}_t^{\pi^\Phi}$ as the baseline, akin to the fully-observable case. □

The following result extends the compatible function approximation theorem in Kakade (2001) to POMDPs.

*Proof of Prop. 6.1.* The proof is identical to Kakade (2001). By first-order condition for optimality, we have

$$2\mathbb{E}_\mu^{\pi^\Phi} \sum_{t=0}^\infty \gamma^t \nabla \ln \pi_t^\Phi(A_t|Z_t) \left( \nabla^\top \ln \pi_t^\Phi(A_t|Z_t)\omega^\star - \mathcal{A}_t^{\pi^\Phi}(\bar{Z}_t) \right) = 2 \left( G_\mu(\Phi)\omega^\star - \nabla_\Phi \mathcal{V}^{\pi^\Phi}(\mu) \right) = 0,$$

which concludes the proof. $\square$

# E  Theoretical Analysis of Rec-NPG

First, we prove structural results for RNNs in the kernel regime, which will be key in the analysis later.

## E.1  Log-Linearization of SoftMax Policies Parameterized by RNNs

The key idea behind the neural tangent kernel (NTK) analysis is linearization around the random initialization. To that end, let

$$F_t^{\mathsf{Lin}}(\bar{z}_t; \Theta) := \langle \nabla F_t(\bar{z}_t; \Theta(0)), \Theta - \Theta(0) \rangle, \tag{61}$$

for any $\Theta \in \mathbb{R}^{m(d+1)}$. We define the log-linearized policy as follows:

$$\tilde{\pi}_t^\Phi(a|z_t) := \frac{\exp(F_t^{\mathsf{Lin}}(z_t, a; \Phi))}{\sum_{a' \in \mathbb{A}} \exp(F_t^{\mathsf{Lin}}(z_t, a'; \Phi))}, \ t \in \mathbb{N}. \tag{62}$$

The first result bounds the Kullback-Leibler divergence between $\pi_t^\Phi$ and its log-linearized version $\tilde{\pi}_t^\Phi$. In the case of FNNs with ReLU activation functions, a similar result was presented in Cayci et al. (2024b). The following result extends this idea to (i) RNNs, and (ii) smooth activation functions.

**Proposition E.1** (Log-linearization error). *For any $t \in \mathbb{N}$ and $(z_t, a) \in (\mathbb{Y} \times \mathbb{A})^{t+1}$, we have*

$$\sup_{(z_t, a) \in (\mathbb{Y} \times \mathbb{A})^{t+1}} \left| \ln \frac{\tilde{\pi}_t^\Phi(a|z_t)}{\pi_t^\Phi(a|z_t)} \right| \leq \frac{6}{\sqrt{m}} \left( \Lambda_t^2 \varrho_2 + \chi_t \varrho_1 \right) \|\Phi - \Phi(0)\|_2^2, \tag{63}$$

*for any $t \in \mathbb{N}$. Consequently, we have $\pi_t(\cdot|z_t) \ll \tilde{\pi}_t(\cdot|z_t)$ and $\tilde{\pi}_t(\cdot|z_t) \ll \pi_t(\cdot|z_t)$, and*

$$\max \left\{ \mathscr{D}_{\mathsf{KL}}(\pi_t^\Phi(\cdot|z_t)\|\tilde{\pi}_t^\Phi(\cdot|z_t)), \mathcal{D}_{\mathsf{KL}}(\tilde{\pi}_t^\Phi(\cdot|z_t)\|\pi_t^\Phi(\cdot|z_t)) \right\} \leq \frac{6}{\sqrt{m}} \left( \Lambda_t^2 \varrho_2 + \chi_t \varrho_1 \right) \|\Phi - \Phi(0)\|_2^2, \tag{64}$$

*for all $z_t \in (\mathbb{Y} \times \mathbb{A})^{t+1}$ and $t \in \mathbb{N}$.*

*Proof.* Fix $(z_t, a) \in (\mathbb{Y} \times \mathbb{A})^{t+1}$. By the log-sum inequality Cover & Thomas (2006), we have

$$\ln \frac{\sum_a \exp(F_t^{\mathsf{Lin}}(z_t, a; \Phi))}{\sum_a \exp(F_t(z_t, a; \Phi))} \leq \sum_{a \in \mathbb{A}} \tilde{\pi}_t^\Phi(a|z_t) \left( F_t^{\mathsf{Lin}}(z_t, a; \Phi) - F_t(z_t, a; \Phi) \right).$$

Using the same argument, we obtain

$$\left| \ln \frac{\sum_a \exp(F_t^{\mathsf{Lin}}(z_t, a; \Phi))}{\sum_a \exp(F_t(z_t, a; \Phi))} \right| \leq \sum_{a \in \mathbb{A}} \left( \tilde{\pi}_t^\Phi(a|z_t) + \pi_t^\Phi(a|z_t) \right) \cdot \left| F_t^{\mathsf{Lin}}(z_t, a; \Phi) - F_t(z_t, a; \Phi) \right|. \tag{65}$$

Thus, we have

$$\left| \ln \frac{\tilde{\pi}_t^\Phi(a|z_t)}{\pi_t^\Phi(a|z_t)} \right| \leq (1 + \tilde{\pi}_t^\Phi(a|z_t) + \pi_t^\Phi(a|z_t)) \left| F_t^{\mathsf{Lin}}(z_t, a; \Phi) - F_t(z_t, a; \Phi) \right|.$$

By using Lemma B.1, we have $\sup_{\bar{z}_t \in (\mathbb{Y} \times \mathbb{A})^{t+1}} \left| F_t^{\mathsf{Lin}}(\bar{z}'_t; \Phi) - F_t(\bar{z}'_t; \Phi) \right| \leq \frac{2}{\sqrt{m}}(\Lambda_t^2 \varrho_2 + \chi_t \varrho_1) \|\Phi - \Phi(0)\|_2^2$. By using the last two inequalities together, and noting that $1 + \tilde{\pi}_t^\Phi(a|z_t) + \pi_t^\Phi(a|z_t) \leq 3$, we conclude that

$$\left| \ln \frac{\tilde{\pi}_t^\Phi(a|z_t)}{\pi_t^\Phi(a|z_t)} \right| \leq \frac{6}{\sqrt{m}}(\Lambda_t^2 \varrho_2 + \chi_t \varrho_1) \|\Phi - \Phi(0)\|_2^2.$$

Since the right-hand side of the above inequality is independent of $(z_t, a)$, we deduce that the result holds for all $(z_t, a)$, thus concluding the proof. $\qquad \square$

The following result will be important in establishing the Lyapunov drift analysis of Rec-NPG.

**Proposition E.2** (Smoothness of $\ln \tilde{\pi}_t^\Phi(a|z_t)$). *For any $t \in \mathbb{N}$, we have*

$$\sup_{(z_t, a) \in (\mathbb{Y} \times \mathbb{A})^{t+1}} \|\nabla \ln \tilde{\pi}_t^\Phi(a|z_t) - \nabla \ln \tilde{\pi}_t^{\Phi'}(a|z_t)\|_2 \leq L_t^2 \|\Phi - \Phi'\|_2,$$

*for any $\Phi, \Phi' \in \mathbb{R}^{m(d+1)}$.*

*Proof.* Consider a general log-linear parameterization

$$p_\theta(x) \propto \exp(\phi_x^\top \theta), \ x \in \mathbf{X}.$$

Then, if $\sup_{x \in \mathbf{X}} \|\phi_x\|_2 \leq B < \infty$, then $\theta \mapsto \ln p_\theta(x)$ has $B^2$-Lipschitz continuous gradients for each $x \in \mathbf{X}$ Agarwal et al. (2020). The remaining part is to prove a uniform upper bound for $\|\nabla_\Phi F_t(\bar{z}_t; \Phi(0))\|_2$. To that end, notice that

$$\nabla_{\Phi_i} F_t(\bar{z}_t; \Phi(0)) = \frac{1}{\sqrt{m}} c_i \nabla H_t^{(i)}(\bar{z}_t; \Phi(0)), \ \bar{z}_t \in (\mathbb{Y} \times \mathbb{A})^{t+1}, i \in [m].$$

From the local Lipschitz continuity result in Lemma B.1, we have $\sup_{\bar{z}_t : \max_{j \leq t} \|(y_j, a_j)\|_2 \leq 1} \|\nabla_{\Phi_i} H_t^{(i)}(\bar{z}_t; \Phi(0))\|_2 \leq L_t$ for any $i \in [m]$. Thus, for any $\bar{z}_t$, we have

$$\|\nabla_\Phi F_t(\bar{z}_t; \Phi(0))\|_2^2 = \frac{1}{m} \sum_{i=1}^m \|\nabla_{\Phi_i} H_t^{(i)}(\bar{z}_t; \Phi(0))\|_2^2 \leq L_t^2. \tag{66}$$

$\square$

## E.2 Theoretical Analysis of Rec-NPG

For any $\pi \in \Pi_{\mathsf{NM}}$, we define the potential function as

$$\mathscr{L}(\pi) := \mathbb{E}_\mu^{\pi^\star} \left[ \sum_{t=0}^{T-1} \gamma^t \mathscr{D}_{\mathsf{KL}} \left( \pi_t^\star(\cdot|Z_t) \| \pi_t(\cdot|Z_t) \right) \right]. \tag{67}$$

Then, we have the following drift inequality.

**Proposition E.3** (Drift inequality). *For any $n \in \mathbb{N}$, the drift can be bounded as follows:*

$$\mathscr{L}(\pi^{\Phi(n+1)}) - \mathscr{L}(\pi^{\Phi(n)}) \leq -\eta_{\mathsf{npg}}(\mathcal{V}^{\pi^\star}(\mu) - \mathcal{V}^{\pi^{\Phi(n)}}(\mu)) \underbrace{-\eta_{\mathsf{npg}} \mathbb{E}_\mu^{\pi^\star} \left[ \sum_{t=0}^{T-1} \gamma^t \left( \nabla^\top \ln \pi_t^{\Phi(n)}(A_t|Z_t)\omega_n - \mathcal{A}_t^{\pi^{\Phi(n)}}(\bar{Z}_t) \right) \right]}_{\text{①}}$$

$$\underbrace{+ \eta_{\mathsf{npg}} \mathbb{E}_\mu^{\pi^\star} \sum_{t=T}^{\infty} \gamma^t \mathcal{A}_t^{\pi^{\Phi(n)}}(\bar{Z}_t)}_{\text{②}} \underbrace{- \eta_{\mathsf{npg}} \mathbb{E}_\mu^{\pi^\star} \sum_{t=0}^{T-1} \gamma^t \left( \nabla \ln \tilde{\pi}_t^{\Phi(n)}(A_t|Z_t) - \nabla \ln \pi_t^{\Phi(n)}(A_t|Z_t) \right)^\top \omega_n}_{\text{③}}$$

$$+ \frac{1}{2} \eta_{\mathsf{npg}}^2 \|\rho\|_2^2 \sum_{t=0}^{T-1} \gamma^t L_t^2 + \frac{12\|\rho\|_2^2}{\sqrt{m}} \sum_{t=0}^{T-1} \gamma^t (\Lambda_t^2 \varrho_2 + \chi_t \varrho_1).$$

*Proof.* First, note that the drift can be expressed as

$$\mathscr{L}(\pi^{\Phi(n+1)}) - \mathscr{L}(\pi^{\Phi(n)}) = \mathbb{E}_\mu^{\pi^\star} \sum_{t=0}^{T-1} \gamma^t \sum_{a \in \mathbb{A}} \pi_t^\star(A_t|Z_t) \ln \frac{\pi_t^{\Phi(n)}(A_t|Z_t)}{\pi_t^{\Phi(n+1)}(A_t|Z_t)}.$$

Then, with a log-linear transformation,

$$\mathscr{L}(\pi^{\Phi(n+1)}) - \mathscr{L}(\pi^{\Phi(n)}) = \mathbb{E}_\mu^{\pi^\star} \sum_{t=0}^{T-1} \gamma^t \sum_{a \in \mathbb{A}} \pi_t^\star(A_t|Z_t) \left( \ln \frac{\tilde{\pi}_t^{\Phi(n)}(A_t|Z_t)}{\tilde{\pi}_t^{\Phi(n+1)}(A_t|Z_t)} + \ln \frac{\pi_t^{\Phi(n)}(A_t|Z_t)}{\tilde{\pi}_t^{\Phi(n)}(A_t|Z_t)} + \ln \frac{\tilde{\pi}_t^{\Phi(n+1)}(A_t|Z_t)}{\pi_t^{\Phi(n+1)}(A_t|Z_t)} \right).$$

By using the log-linearization bound in Prop. E.1 twice in the above inequality, we obtain

$$\mathscr{L}(\pi^{\Phi(n+1)}) - \mathscr{L}(\pi^{\Phi(n)}) \le \mathbb{E}_\mu^{\pi^\star} \sum_{t=0}^{T-1} \gamma^t \sum_{a \in \mathbb{A}} \pi_t^\star(A_t|Z_t) \ln \frac{\tilde{\pi}_t^{\Phi(n)}(A_t|Z_t)}{\tilde{\pi}_t^{\Phi(n+1)}(A_t|Z_t)} + \frac{12}{\sqrt{m}} \sum_{t=0}^{T-1} \gamma^t (\Lambda_t^2 \varrho_2 + \chi_t \varrho_1) \|\rho\|_2^2. \quad (68)$$

By the smoothness result in Prop. E.2, we have

$$\left| \ln \tilde{\pi}_t^{\Phi(n+1)}(a_t|z_t) - \ln \tilde{\pi}_t^{\Phi(n)}(a_t|z_t) - \nabla \ln \tilde{\pi}_t^{\Phi(n)}(a_t|z_t)(\Phi(n+1) - \Phi(n)) \right| \le \frac{1}{2} L_t^4 \|\Phi(n+1) - \Phi(n)\|_2^2.$$

Thus, we obtain

$$-\eta_{\mathsf{npg}}^2 L_t^4 \|\rho\|_2^2 \le -\eta_{\mathsf{npg}}^2 L_t^4 \|\omega_n\|_2^2 \le -\ln \frac{\tilde{\pi}_t^{\Phi(n)}(a_t|z_t)}{\tilde{\pi}_t^{\Phi(n+1)}(a_t|z_t)} - \eta_{\mathsf{npg}} \nabla^\top \ln \tilde{\pi}_t^{\Phi(n)}(a_t|z_t)\omega_n,$$

because of the max-norm gradient clipping that yields $\|\omega_n\|_2 \le \|\rho\|_2$ and $\Phi(n+1) = \Phi(n) + \eta_{\mathsf{npg}}\omega_n$ for any $n \in \mathbb{N}$. Using this in equation 68, we get

$$\mathscr{L}(\pi^{\Phi(n+1)}) - \mathscr{L}(\pi^{\Phi(n)}) \le -\eta_{\mathsf{npg}} \mathbb{E}_\mu^{\pi^\star} \sum_{t=0}^{T-1} \gamma^t \nabla^\top \ln \tilde{\pi}_t^{\Phi(n)}(a_t|z_t)\omega_n + \frac{12}{\sqrt{m}} \sum_{t=0}^{T-1} \gamma^t (\Lambda_t^2 \varrho_2 + \chi_t \varrho_1) \|\rho\|_2^2 + \frac{1}{2} \eta_{\mathsf{npg}}^2 L_t^4 \|\rho\|_2^2.$$
$$(69)$$

An important technical result that will be useful in our analysis is the *pathwise* performance difference lemma, which was originally developed in Kakade & Langford (2002) for fully-observable MDPs.

**Lemma E.4** (Pathwise Performance Difference Lemma). *Let $\Phi, \Phi' \in \mathbb{R}^{m(d+1)}$ be two parameters. Then, we have*

$$\mathcal{V}^{\pi^{\Phi'}}(\mu) - \mathcal{V}^{\pi^\Phi}(\mu) = \mathbb{E}_\mu^{\pi^{\Phi'}} \sum_{t=0}^\infty \gamma^t \mathcal{A}_t^{\pi^\Phi}(Z_t, A_t).$$

The proof of Lemma E.4 is an extension of Agarwal et al. (2020) to non-stationary policies, and can be found at the end of this subsection.

Using Lemma E.4 in equation 69, we obtain

$$\mathscr{L}(\pi^{\Phi(n+1)}) - \mathscr{L}(\pi^{\Phi(n)}) \le -\eta_{\mathsf{npg}} (\mathcal{V}^{\pi^\star}(\mu) - \mathcal{V}^{\pi^{\Phi(n)}}(\mu)) - \eta_{\mathsf{npg}} \mathbb{E}_\mu^{\pi^\star} \sum_{t=0}^{T-1} \gamma^t \left( \nabla^\top \ln \tilde{\pi}_t^{\Phi(n)}(a_t|z_t)\omega_n - \mathcal{A}_t^{\pi^{\Phi(n)}}(\bar{Z}_t) \right)$$

$$+ \eta_{\mathsf{npg}} \mathbb{E}_\mu^{\pi^\star} \sum_{t=T}^\infty \mathcal{A}_t^{\pi^{\Phi(n)}}(\bar{Z}_t) + \frac{12}{\sqrt{m}} \sum_{t=0}^{T-1} \gamma^t (\Lambda_t^2 \varrho_2 + \chi_t \varrho_1) \|\rho\|_2^2 + \frac{1}{2} \eta_{\mathsf{npg}}^2 L_t^4 \|\rho\|_2^2. \quad (70)$$

Finally, we replace the term $\nabla \ln \tilde{\pi}_t^{\Phi(n)}(a_t|z_t)$ with $\nabla \ln \pi_t^{\Phi(n)}(a_t|z_t)$ by including the corresponding error term, and conclude the proof by considering the telescoping sum, and noting that $\mathscr{L}(\pi^{\Phi(0)}) = \log|\mathbb{A}|$ since $F_t(\cdot; \Phi(0)) = 0$ by symmetric initialization. $\qquad \square$

*Proof of Theorem 6.3.* We prove Theorem 6.3 by bounding the numbered terms in Prop. E.3.

**Bounding ① in Prop. E.3.** Recall that $p_T(\gamma) = \sum_{t<T} \gamma^t$. Then, by using Jensen's inequality,

$$\mathbb{E}_\mu^{\pi^\star} \sum_{t=0}^{T-1} \gamma^t \left( \nabla^\top \ln \pi_t^{\Phi(n)}(A_t|Z_t)\omega_n - \mathcal{A}_t^{\pi^{\Phi(n)}}(\bar{Z}_t) \right) \leq \sqrt{p_T(\gamma)\mathbb{E}_\mu^{\pi^\star} \sum_{t=0}^{T-1} \gamma^t \left| \nabla^\top \ln \pi_t^{\Phi(n)}(A_t|Z_t)\omega_n - \mathcal{A}_t^{\pi^{\Phi(n)}}(\bar{Z}_t) \right|^2},$$

$$=: \sqrt{p_T(\gamma)}\sqrt{\kappa\varepsilon_{\mathsf{cfa}}^T(\Phi(n),\omega_n)},$$

where $\kappa$ yields a change-of-measure argument from $P_T^{\pi^\star,\mu}$ to $P_T^{\pi^{\Phi(n)},\mu}$.

**Bounding ② in Prop. E.3.** $\sup_{s,a} |r(s,a)| \leq r_\infty$, therefore $|\mathcal{A}_t^\pi(\bar{z}_t)| \leq \frac{2r_\infty}{1-\gamma}$ for any $t \in \mathbb{N}, \bar{z}_t \in (\mathbb{Y} \times \mathbb{A})^{t+1}$, and $\pi \in \Pi_{\mathsf{NM}}$.

**Bounding ③ in Prop. E.3.** For any $t \in \mathbb{N}$, Cauchy-Schwarz inequality implies

$$\left( \nabla \ln \tilde{\pi}_t^{\Phi(n)}(a_t|z_t) - \nabla \ln \pi_t^{\Phi(n)}(a_t|z_t) \right)^\top \omega_n \leq \| \nabla \ln \tilde{\pi}_t^{\Phi(n)}(a_t|z_t) - \nabla \ln \pi_t^{\Phi(n)}(a_t|z_t) \|_2 \|\rho\|_2.$$

Recall that

$$\nabla \ln \tilde{\pi}_t^\Phi(a_t|z_t) = \nabla F_t(z_t, a_t; \Phi(0)) - \sum_{a'} \tilde{\pi}_t^\Phi(a'|z_t)\nabla F_t(z_t, a'; \Phi(0)),$$

$$\nabla \ln \pi_t^\Phi(a_t|z_t) = \nabla F_t(z_t, a_t; \Phi) - \sum_{a'} \pi_t^\Phi(a'|z_t)\nabla F_t(z_t, a'; \Phi).$$

First, from local $\beta_t$-Lipschitzness of $\Phi_i \mapsto \nabla H_t^{(i)}(\bar{z}_t; \Phi_i)$ for $\Phi \in \Omega_{\rho,m}$ by Lemma B.1, we have

$$\|\nabla_{\Phi_i} F_t(\bar{z}_t; \Phi(n)) - \nabla_{\Phi_i} F_t(\bar{z}_t; \Phi(0))\|_2 = \frac{1}{\sqrt{m}}\|\nabla_{\Phi_i} H_t^{(i)}(\bar{z}_t; \Phi_i(n)) - \nabla_{\Phi_i} H_t^{(i)}(\bar{z}_t; \Phi_i(0))\|_2,$$

$$\leq \frac{\beta_t\|\rho\|_2}{m},$$

for any $n \in \mathbb{N}$ since $\max_i \|\Phi_i(n) - \Phi_i(0)\|_2 \leq \frac{\|\rho\|_2}{\sqrt{m}}$ by max-norm projection. Thus,

$$\|\nabla_\Phi F_t(\bar{z}_t; \Phi(n)) - \nabla_\Phi F_t(\bar{z}_t; \Phi(0))\|_2 \leq \frac{\beta_t\|\rho\|_2}{\sqrt{m}}, \quad t \in \mathbb{N}. \tag{71}$$

Thus,

$$\|\nabla \ln \tilde{\pi}_t^{\Phi(n)}(a_t|z_t) - \nabla \ln \pi_t^{\Phi(n)}(a_t|z_t)\|_2 \leq \frac{\beta_t\|\rho\|_2}{\sqrt{m}} + \sum_a |\pi_t^{\Phi(n)}(a|z_t) - \tilde{\pi}_t^{\Phi(n)}(a|z_t)|\|\nabla F_t(\bar{z}_t; \Phi(0))\|_2$$

$$+ \sum_a \pi_t^{\Phi(n)}(a|z_t)\|\nabla F_t(z_t, a; \Phi(n)) - \nabla F_t(z_t, a; \Phi(0))\|_2.$$

From equation 66, we have

$$\|\nabla \ln \tilde{\pi}_t^{\Phi(n)}(a_t|z_t) - \nabla \ln \pi_t^{\Phi(n)}(a_t|z_t)\|_2 \leq \frac{2\beta_t\|\rho\|_2}{\sqrt{m}} + 2L_t\mathscr{D}_{\mathsf{TV}}\left( \pi_t^{\Phi(n)}(\cdot|z_t)\|\tilde{\pi}_t^{\Phi(n)}(\cdot|z_t) \right),$$

where $\mathscr{D}_{\mathsf{TV}}$ denotes the total-variation distance between two probability measures. By Pinsker's inequality Cover & Thomas (2006), we obtain

$$\|\nabla \ln \tilde{\pi}_t^{\Phi(n)}(a_t|z_t) - \nabla \ln \pi_t^{\Phi(n)}(a_t|z_t)\|_2 \leq \frac{2\beta_t\|\rho\|_2}{\sqrt{m}} + \sqrt{2}L_t\sqrt{\mathscr{D}_{\mathsf{KL}}\left( \pi_t^{\Phi(n)}(\cdot|z_t)\|\tilde{\pi}_t^{\Phi(n)}(\cdot|z_t) \right)}. \tag{72}$$

By the log-linearization result in Prop. E.1, we have

$$\|\nabla \ln \tilde{\pi}_t^{\Phi(n)}(a_t|z_t) - \nabla \ln \pi_t^{\Phi(n)}(a_t|z_t)\|_2 \leq \frac{2\beta_t \|\rho\|_2}{\sqrt{m}} + \sqrt{12} L_t \|\rho\|_2 \sqrt{\frac{\Lambda_t^2 \varrho_2 + \chi_t \varrho_1}{\sqrt{m}}}. \tag{73}$$

Thus, we have

$$\left( \nabla \ln \tilde{\pi}_t^{\Phi(n)}(a_t|z_t) - \nabla \ln \pi_t^{\Phi(n)}(a_t|z_t) \right)^\top \omega_n \leq \|\rho\|_2^2 \left( \frac{2\beta_t}{\sqrt{m}} + \sqrt{12} L_t \frac{\sqrt{\Lambda_t \varrho_2 + \chi_t \varrho_1}}{m^{1/4}} \right).$$

$\square$

*Proof of Lemma E.4.* For any $y_0 \in \mathbb{Y}$, we have:

$$\mathcal{V}^{\pi'}(y_0) - \mathcal{V}^\pi(y_0) = \mathbb{E}_\mu^{\pi'} \left[ \sum_{t=0}^\infty \gamma^t r_t \Big| Z_0 = y_0 \right] - \mathcal{V}^\pi(y_0),$$

$$= \mathbb{E}_\mu^{\pi'} \left[ \sum_{t=0}^\infty \gamma^t \left( r_t + \mathcal{V}_t^\pi(Z_t) - \mathcal{V}_t^\pi(Z_t) \right) \Big| Z_0 = y_0 \right] - \mathcal{V}^\pi(y_0),$$

$$= \mathbb{E}_\mu^{\pi'} \left[ \sum_{t=0}^\infty \gamma^t (r_t + \gamma \mathcal{V}_{t+1}^\pi(Z_{t+1}) - \mathcal{V}_t^\pi(Z_t) \Big| Z_0 = y_0 \right],$$

where $r_t = r(S_t, A_t)$ and the last identity holds since

$$\sum_{t=0}^\infty \gamma^t \mathcal{V}_t^\pi(z_t) = \mathcal{V}_0^\pi(z_0) + \gamma \sum_{t=0}^\infty \gamma^t \mathcal{V}_{t+1}^\pi(z_{t+1}).$$

Then, letting $r_t = r(s_t, a_t)$ and by using law of iterated expectations,

$$\mathcal{V}^{\pi'}(y_0) - \mathcal{V}^\pi(y_0) = \mathbb{E}_\mu^{\pi'} \left[ \sum_{t=0}^\infty \gamma^t \left( \mathbb{E}^{\pi'}[r_t + \gamma \mathcal{V}_{t+1}^\pi(Z_{t+1})|\bar{Z}_t, S_t] - \mathcal{V}_t^\pi(Z_t) \right) \Big| Z_0 = y_0 \right], \tag{74}$$

which holds because

$$\mathbb{E}^{\pi'}[r_t + \gamma \mathcal{V}^\pi(Z_{t+1})|\bar{Z}_t] = \mathbb{E}^{\pi'}[r_t + \gamma \mathcal{V}^\pi(Z_{t+1})|\bar{Z}_t, Z_0].$$

The conditional expectation of $r_t + \gamma \mathcal{V}_{t+1}^\pi$ given $\{\bar{Z}_t = \bar{z}_t\}$ is independent of $\pi'$:

$$\mathbb{E}^{\pi'}[r_t + \gamma \mathcal{V}^\pi(Z_{t+1})|\bar{Z}_t] = \sum_{s \in \mathbb{S}} b_t(s) \mathbb{E}^{\pi'}[r_t + \gamma \mathcal{V}_{t+1}^\pi(Z_{t+1})|\bar{Z}_t = \bar{z}_t, S_t = s],$$

$$= \sum_{s_t, s_{t+1} \in \mathbb{S}} \sum_{y \in \mathbb{Y}} b_t(s_t) \left( r(s_t, A_t) + \gamma \mathcal{P}(s_{t+1}|s_t, A_t) \phi(y|s_{t+1}) \mathcal{V}_{t+1}^\pi(Z_t, y_{t+1}) \right),$$

$$= \mathbb{E}[r_t + \gamma \mathcal{V}_{t+1}^\pi(Z_{t+1})|\bar{Z}_t = \bar{z}_t],$$

based on Prop. D.1. We also know from Prop. B.3 that

$$\mathbb{E}^{\pi'}[r_t + \gamma \mathcal{V}_{t+1}^\pi(Z_{t+1})|\bar{Z}_t = \bar{z}_t] = \mathbb{E}[r_t + \gamma \mathcal{V}_{t+1}^\pi(Z_{t+1})|\bar{Z}_t = \bar{z}_t] = \mathcal{Q}_t^\pi(\bar{z}_t).$$

Using the above identity in equation 74, we obtain

$$\mathcal{V}^{\pi'}(y_0) - \mathcal{V}^\pi(y_0) = \mathbb{E}_\mu^{\pi'} \left[ \sum_{t=0}^\infty \gamma^t \left( \mathcal{Q}_t^\pi(\bar{Z}_t) - \mathcal{V}^\pi(Z_t) \right) \Big| Z_0 = y_0 \right], \tag{75}$$

which concludes the proof. $\square$

*Proof of Prop. 6.6.* For any $\omega$, we have

$$\ell_T(\omega; \Phi(n), \mathcal{Q}^{\pi^{\Phi(n)}}) \le 2\ell_T(\omega; \Phi(n), \hat{\mathcal{Q}}^{(n)}) + 2\sum_{t=0}^{\infty} \gamma^t (\mathcal{A}_t^{\pi^{\Phi(n)}}(Z_t, A_t) - \hat{\mathcal{A}}_t^{(n)}(Z_t, A_t))^2. \tag{76}$$

Let $\mathcal{G}_n := \sigma(\Phi(k), k \le n)$ and $\mathcal{H}_n := \sigma(\bar{\Theta}^{(n)}, \Phi(k), k \le n)$. Then, since

$$\varepsilon_{\mathsf{sgd},n} = \mathbb{E}[\ell_T(\omega_n; \Phi(n), \hat{\mathcal{Q}}^{(n)}) | \mathcal{H}_n] - \inf_{\omega \in \mathcal{B}_{2,\infty}^{(m)}(0,\rho)} \mathbb{E}[\ell_T(\omega; \Phi(n), \hat{\mathcal{Q}}^{(n)}) | \mathcal{H}_n],$$

we obtain

$$\mathbb{E}[\ell_T(\omega_n; \Phi(n), \mathcal{Q}^{\pi^{\Phi(n)}}) | \mathcal{H}_n] \le 2\mathbb{E}\Big[\inf_{\omega} \mathbb{E}[\ell_T(\omega; \Phi(n), \hat{\mathcal{Q}}^{(n)}) | \mathcal{H}_n]\Big|\mathcal{G}_n\Big] + 2(\varepsilon_{\mathsf{td},n} + \varepsilon_{\mathsf{sgd},n}), \tag{77}$$

which uses the fact that $Var(X|\mathcal{G}_n) \le \mathbb{E}[|X|^2|\mathcal{G}_n]$ for any square-integrable $X$. We also have

$$\inf_{\omega} \mathbb{E}[\ell_T(\omega; \Phi(n), \hat{\mathcal{Q}}^{(n)}) | \mathcal{H}_n] \le 2\inf_{\omega} \mathbb{E}[\ell_T(\omega; \Phi(n), \mathcal{Q}^{\pi^{\Phi(n)}}) | \mathcal{H}_n] + 2\sum_{t=0}^{\infty} \gamma^t (\mathcal{A}_t^{\pi^{\Phi(n)}}(Z_t, A_t) - \hat{\mathcal{A}}_t^{(n)}(Z_t, A_t))^2, \tag{78}$$

which further implies that

$$\mathbb{E}[\inf_{\omega} \mathbb{E}[\ell_T(\omega; \Phi(n), \hat{\mathcal{Q}}^{(n)}) | \mathcal{H}_n] | \mathcal{G}_n] \le 2\mathbb{E}[\inf_{\omega} \mathbb{E}[\ell_T(\omega; \Phi(n), \mathcal{Q}^{\pi^{\Phi(n)}}) | \mathcal{H}_n] | \mathcal{G}_n] + 2\varepsilon_{\mathsf{td},n}.$$

Thus,

$$\mathbb{E}[\ell_T(\omega_n; \Phi(n), \mathcal{Q}^{\pi^{\Phi(n)}}) | \mathcal{H}_n] \le 4\mathbb{E}\Big[\inf_{\omega} \mathbb{E}[\ell_T(\omega; \Phi(n), \mathcal{Q}^{\pi^{\Phi(n)}}) | \mathcal{H}_n]\Big|\mathcal{G}_n\Big] + 6\varepsilon_{\mathsf{td},n} + 2\varepsilon_{\mathsf{sgd},n}. \tag{79}$$

For any $\omega \in \mathcal{B}_{2,\infty}^{(m)}(0,\rho)$,

$$\mathbb{E}[\ell_T(\omega; \Phi(n), \mathcal{Q}^{\pi^{\Phi(n)}}) | \mathcal{H}_n] \le \mathbb{E}[\sum_{t<T} \gamma^t (\nabla_\Phi^\top F_t(\bar{Z}_t; \Phi(n))\omega - \mathcal{Q}_t^{\pi^{\Phi(n)}}(\bar{Z}_t))^2 | \mathcal{H}_n],$$

$$\le 2\mathbb{E}[\sum_{t<T} \gamma^t (\nabla_\Phi^\top F_t(\bar{Z}_t; \Phi(0))\omega - \mathcal{Q}_t^{\pi^{\Phi(n)}}(\bar{Z}_t))^2 + (\nabla F_t(\bar{Z}_t; \Phi(n)) - \nabla F_t(\bar{Z}_t; \Phi(0))^\top \omega)^2 | \mathcal{H}_n],$$

which implies that

$$\inf_{\omega} \mathbb{E}[\ell_T(\omega; \Phi(n), \mathcal{Q}^{\pi^{\Phi(n)}}) | \mathcal{H}_n] \le 2\varepsilon_{\mathsf{app},n} + 2\|\rho\|_2^2 \mathbb{E}[\sum_{t<T} \gamma^t \|\nabla F_t(\bar{Z}_t; \Phi(n)) - \nabla F_t(\bar{Z}_t; \Phi(0))\|_2^2 | \mathcal{H}_n],$$

$$\le 2\varepsilon_{\mathsf{app},n} + \frac{2\|\rho\|_2^4}{m} \sum_{t<T} \gamma^t \beta_t^2,$$

using equation 71. Hence,

$$\mathbb{E}[\ell_T(\omega_n; \Phi(n), \mathcal{Q}^{\pi^{\Phi(n)}}) | \mathcal{H}_n] \le \frac{8\|\rho\|_2^4}{m} \sum_{t<T} \gamma^t \beta_t^2 + 8\varepsilon_{\mathsf{app},n} + 6\varepsilon_{\mathsf{td},n} + 2\varepsilon_{\mathsf{sgd},n},$$

concluding the proof. $\qquad\square$

*Proof of Prop. 6.8.* Under Assumption 6.7, consider $f_t^{(j)}(\bar{z}_t) := \mathbb{E}[\psi_t^\top(\bar{z}_t; \phi_0)\boldsymbol{v}^{(j)}(\phi_0)]$ for $\boldsymbol{v}^{(j)} \in \mathscr{H}_{\mathcal{J},\nu}$. Let

$$\omega_i^{(j)} := \frac{1}{\sqrt{m}} c_i \boldsymbol{v}^{(j)}(\Phi_i(0)), \ \ i = 1, 2, \ldots, m, \tag{80}$$

for any $j \in \mathcal{J}$. Since $\|\omega^{(j)}\|_2 \le \|\nu\|_2$ and $\rho \succeq \nu$, we have

$$\inf_{\omega \in \mathcal{B}_{2,\infty}^{(m)}(0,\rho)} \Big|\nabla^\top F_t(\bar{z}_t; \Phi(0))\omega - f_t^{(j)}(\bar{z}_t)\Big| \le \Big|\nabla^\top F_t(\bar{z}_t; \Phi(0))\omega^{(j)} - f_t^{(j)}(\bar{z}_t)\Big|. \tag{81}$$

Thus, we aim to find a uniform upper bound for the second term over $j \in \mathcal{J}$. For each $\bar{z}_t$, we have

$$\nabla^\top F_t(\bar{z}_t; \Phi(0))\omega^{(j)} = \frac{1}{m}\sum_{i=1}^m \nabla_{\Phi_i}^\top H_t^{(i)}(\bar{z}_t; \Phi_i(0))\boldsymbol{v}^{(j)}(\Phi_i(0)),$$

thus $\mathbb{E}[\nabla^\top F_t(\bar{z}_t; \Phi(0))\omega^{(j)}] = f_t^{(j)}(\bar{z}_t)$. Furthermore, from Lemma B.1, since $\Phi(0) \in \Omega_{\rho,m}$ obviously, we have

$$\max_{1\le i\le m} \|\nabla_{\Phi_i}^\top H_t^{(i)}(\bar{z}_t; \Phi_i(0))\boldsymbol{v}^{(j)}(\Phi_i(0))\|_2 \le L_t\|\nu\|_2 \le L_t\|\rho\|_2, \text{ a.s.}.$$

Thus, by McDiarmid's inequality Mohri et al. (2018), we have with probability at least $1 - \delta$,

$$\sup_{j\in\mathcal{J}}\left|\nabla^\top F_t(\bar{z}_t; \Phi(0))\omega^{(j)} - f_t^{(j)}(\bar{z}_t)\right| \le 2\mathrm{Rad}_m(G_t^{\bar{z}_t}) + L_t\|\rho\|_2\sqrt{\frac{\log(2/\delta)}{m}}, \tag{82}$$

for each $t < T$ and $\bar{z}_t$. By union bound,

$$\sup_{j\in\mathcal{J}}\max_{\bar{z}_t}\left|\nabla^\top F_t(\bar{z}_t; \Phi(0))\omega^{(j)} - f_t^{(j)}(\bar{z}_t)\right| \le 2\max_{\bar{z}_t}\mathrm{Rad}_m(G_t^{\bar{z}_t}) + L_t\|\rho\|_2\sqrt{\frac{\log(2T|\mathbb{Y}\times\mathbb{A}|^{t+1}/\delta)}{m}}, \tag{83}$$

$$\le 2\max_{0\le t<T}\max_{\bar{z}_t}\mathrm{Rad}_m(G_t^{\bar{z}_t}) + L_T\|\rho\|_2\sqrt{\frac{\log(2T|\mathbb{Y}\times\mathbb{A}|^T/\delta)}{m}}, \tag{84}$$

simultaneously for all $t < T$ with probability $\ge 1 - \delta$. Therefore,

$$\inf_\omega \mathbb{E}_\mu^{\pi^{\Phi(n)}}\sum_{t<T}\gamma^t|\nabla^\top F_t(\bar{Z}_t; \Phi(0))\omega - f_t^{(j)}|^2 \le \mathbb{E}_\mu^{\pi^{\Phi(n)}}\sum_{t<T}\gamma^t\sup_{j\in\mathcal{J}}|\nabla^\top F_t(\bar{Z}_t; \Phi(0))\omega^{(j)} - f_t^{(j)}|^2,$$

$$\le \frac{1}{1-\gamma}\left(2\max_{0\le t<T}\max_{\bar{z}_t}\mathrm{Rad}_m(G_t^{\bar{z}_t}) + L_T\|\rho\|_2\sqrt{\frac{\log(2T|\mathbb{Y}\times\mathbb{A}|^T/\delta)}{m}}\right)^2.$$

$\square$

