# OpenReview forum: "Recurrent Natural Policy Gradient for POMDPs"
_TMLR — Accepted by TMLR_

### Review · Reviewer_q1Mw · 2025-06-26

**Summary Of Contributions:**

This work pioneers the formal analysis of a class of NAC algorithms, which uses RNN to represent both their critic and actor to circumvent POMDP challenges, i.e., a “PO” setup implies the need to keep the policy search space to history-dependent and randomized but such search (without tricks like RNN or other form of memory retention/policy representation) implies exponential memory complexity and non-vanishing optimality gap. Authors obtain error bounds (gap to “optimality”) for both policy evaluation (Rec-TD with max-norm regularization) and policy optimization (Rec-NPG best-iterate), where dependency on memory (T), time/iteration (N), sample (K), and RNN class specification (e.g., $m$, $\varrho$, $\alpha_m$, $\rho$, $\nu$) are characterized explicitly, under some conditions. When these gap vanish (or not) are further discussed; some of which reveal long-term memory as pathological case.

**Audience:**

Yes

**Claims And Evidence:**

Yes

**Requested Changes:**

[1] On the long-term memory pathology case of Theorem 5.4, it seems that “long-term” here (as discussed below Remark 5.6) refers to the choice of RNN ($\alpha$, $\varrho_1$). Should I understand this discussion as a guide to avoid setting my $W(0)$ and activation in the way that $\alpha_m > \frac{1}{\varrho_1}$? Or is this long-term memory setting unavoidable (or even useful) in certain circumstances (say, for RNN training aspect separate from bounding the Rec-TD error)?

[2] While the long-term pathology source seems to come from the choice of RNN (cf. Qn [1]), it seems that the long-term memory illustration in Random-POMDP (Appendix C) refers to a different source, i.e. exploratory policy with small exploration rate. While I can understand that small rate does cause longer historical dependency, I can’t seem to see this in the term $p_T(\alpha_m*\varrho_1)$. Is it related to another term in Eqn (10)? Or should I understand Appendix C separately from the bound in Eqn (10)?

[3] On Theorem 6.3 providing best-iterate error bound, am I right to understand that this bound does not guarantee the algorithm to avoid divergence or collapsing? If true, can Authors provide some intuition on (a) what currently prevents obtaining stronger results, (b) when divergence/collapse might happen during training, how likely or unlikely?

[4] Am I right to understand that with RNN encoding, errors that are originally dependent on memory ($T$) are transferred into RNN approximation error, s.t. “boundedness” now relies on the assumption that RNN class is rich enough to suppress the error? If true, is the specific RNN class in Section 3 chosen to supply such richness? Can Authors discuss the necessity of NTK IndRNN, e.g., whether an alternative class of RNN may cause some terms in the current Error bounds to possibly explode?

[5] To speak form algorithmic perspective, I would want to know how I can utilize the obtained bounds here to guide my hyperparameter (both RNN and RL) choice. While I understand that some Assumptions are difficult to flesh out and some terms rely on the POMDP instance, it would be useful to discuss some examples of when the theoretical bounds can directly translate to training guidelines. Perhaps Authors can expand from the short-term vs long-term memory discussion in Sec 5.2.

**Strengths And Weaknesses:**

The work is sound technically, and its presentation is clear. The RNN-NAC class of algorithms are important in solving POMDP and the provided analyses significantly improve understanding on these algorithms' behavior.

There are no major weaknesses in my view, but addressing some of my clarifying questions (cf. Requested Changes) may help strengthen the work.

---

> ### Author Response · Authors · 2025-07-24
>
> We thank the reviewer for thoroughly reading our paper, and providing very valuable, positive, and constructive feedback, which helped us improve the overall quality of the paper. In the revised paper, we have highlighted the revisions in response to the comments of the reviewers in blue.
>
> Below we provide our responses to the specific points raised by the reviewer.
>
> - **Long-term dependencies.** We thank the reviewer for this important question. In the IndRNN architecture, $|W_{ii}|$ determines how much information from $H_t^{(i)}$ is retained as a form of memory. Under max-norm projection, we have $|W_{ii}(k)| \leq |W_{ii}(0)|+\frac{\rho_w}{\sqrt{m}}$. As such, the projection radius $\rho_w$ determines the memory as it controls $|W_{ii}|$. Now, Theorem 5.4 indicates that the projection radius $\rho=(\rho_w,\rho_u)$ must be sufficiently large so that $\rho \geq \nu$, where $\nu$ is the RKHS norm of $\{\mathcal{Q}_t^\pi:t\in[T]\}$. If $\rho$ is not chosen sufficiently large to ensure $\rho \geq \nu$, then the true Q-functions cannot be learned since the IndRNN is not expressive enough. Therefore, while $\alpha_m$ should be chosen as small as possible, there is a minimum value for that as a function of the RKHS norm (i.e., smoothness) of the Q-functions.
>
> - **Memory in the Random-POMDP example.** We thank the reviewer for this question. In the Random-POMDP example, large $p_{\mathsf{exp}}$ corresponds to i.i.d. (memoryless) exploration, while small $p_{\mathsf{exp}}$ corresponds to choosing the best action in the history so far, requiring a larger memory. As such, large $p_{\mathsf{exp}}$ indicates that Rec-TD can learn the true value functions with only small $\max_i|W_{ii}(k)|$ throughout the iterations $k$, thereby avoiding the exploding gradient problem.
>
> - **Best-iterate bounds for natural policy gradient.** The reviewer is absolutely right as Theorem 6.3 only indicates performance guarantees on the best-iterate rather than last-iterate. This is exactly the same for NPG with function approximation in the case of MDPs (see (Wang et al., 2019; Agarwal et al., 2021)). The main reasons for this challenge are two-fold: (i) without overparameterization, the natural policy gradient is not a pseudo-contraction with respect to the Lyapunov function in Equation (60), and (ii) the natural policy gradient $\Phi\mapsto G_\mu^\dagger(\Phi)\nabla_\Phi \mathcal{V}^{\pi^\Phi}(\mu)$ is not Lipschitz continuous in general. In Euclidean optimization, this is analogous to the absence of (i) strong convexity, and (ii) smoothness, which leads to only best-iterate guarantees under projected-GD. To overcome this and achieve the desirable last-iterate convergence bounds, entropy regularization is used (see (Cayci et al., 2024)). However, this comes at the cost of introducing a bias, as the policy optimization is then performed over the regularized problem rather than the original one. Constructing an example where NPG with function approximation is actually divergent is a very interesting direction for future research.
>
> - **Approximation and impact of $T$.** The function class $\mathscr{F}_T$ corresponds to the RKHS of the neural tangent kernel induced by IndRNNs, and directly extends the reference function classes used in the neural RL literature (Cai et al., 2019; Wang et al., 2019; Liu et al., 2019) as discussed in Remark 3.4. We make the representational assumptions or approximation error calculations with respect to this concrete reference function class. Now, any function sequence in $\mathscr{F}_T$ can be approximated by a finite-width randomly-initialized IndRNN of width $m$, and this approximation error for approximating $\mathscr{F}_T$ using IndRNNs of width $m$ is $\mathcal{O}(c_T/\sqrt{m})$ by Lemma B.2 in our paper and Proposition 3.8 in (Cayci and Eryilmaz, 2025). Here, the coefficient $c_T$ also reflects the impact of the sequence-length $T$ and long-term dependencies, akin to $C_T^{(1)}$ and $C_T^{(2)}$ in Theorem 5.4. Furthermore, Theorem 5.4 proves that $(\mathcal{Q}_t^\pi:t<T)\in\mathscr{F}_T$ can be learned by Rec-TD using a finite-width IndRNN, with finer precision as $m$ increases. We have expanded Section 3.1, and provided additional discussion just before Remark 5.5 to elaborate more on the approximation using this IndRNNs and their NTK RKHS.

---

> ### Author Response · Authors · 2025-07-24
>
> - **Practical implications.** Thank you for your feedback and suggestions. We totally agree with the reviewer that additional discussion on the practical implications of the paper is important. To address the reviewer's comment, we have now expanded Remark 5.5 in the revision to discuss the above practical implications. In the following, we outline some of these implications.
>
>   The analysis in Theorem 5.4 provides several insights into the regularization and hyperparameter selection to stabilize the training of IndRNNs by Rec-TD. On the one hand, it reveals the critical importance of applying a max-norm projection to control $\max_{i\in[m]}|W_{ii}|$ to ensure the stability of IndRNNs for large $T$, and prescribes the choice of the projection radius $\rho$. As such, a practitioner would be well-advised to use domain knowledge or adaptive strategies to adjust the projection parameter. On the other hand, it reveals the impact of the sequence-length $T$, neural network size $m$ and the number of iterations $K$ to achieve a given target error. By characterizing the rates of convergence in each of these parameters, the bound also suggests how to increase these choices in order to improve the performance. Moreover, the bound reveals the impact of long-term dependencies that is discussed in Remark 5.6.

---

### Review · Reviewer_qNLx · 2025-07-07

**Summary Of Contributions:**

The paper analyzes some of the theoretical properties for combining diagonal recurrent architectures with natural policy gradient methods to solve POMDPs.
The paper asks three important questions that are nicely built on top of each other. Starting with the policy evaluation setting, the paper shows that using IndRNNs with TD for policy evaluation (Rec-TD) can achieve a near-optimal solution, and it characterizes some issues that appear when dealing with long-term memory. Then, moving to the control problem, the paper discusses the use of IndRNNs for policy parametrization (Rec-NPG). Finally, the paper then analyses the combination of Rec-TD and Rec-NPG and shows that there’s an exponential growth in the resources required when dealing with long-term memory.

**Audience:**

Yes

**Broader Impact Concerns:**

No concerns.

**Claims And Evidence:**

Yes

**Requested Changes:**

- A brief discussion on how this analysis could be extended to a Dense recurrent layer would be a nice addition. I am not sure where the diagonal properties of the IndRNNs were used in the analysis.
- Changing the matrix-vector multiplications with W to element-wise multiplications would make the maths more readable.
- Remove the multi-step TD claims.
- Include a limitation section discussing which of the assumptions made are reasonable from a practical view and which are more constraining.
- Previous work [1] has analyzed the exploding gradient for diagonal recurrence, proposing that a simple normalization on the inputs mitigates this issue. I wonder if having such normalization would help get rid of the exploding issue in the analysis. Maybe a brief discussion on how this affects the bounds of the critic error is useful.



[1] Zucchet, Nicolas, and Antonio Orvieto. "Recurrent neural networks: vanishing and exploding gradients are not the end of the story." Advances in Neural Information Processing Systems 37 (2024): 139402-139443.

**Strengths And Weaknesses:**

### Strengths
- The finite-time bounds result for Rec-TD is novel and valuable as it quantifies the error in terms of the network width, number of iterations, and memory horizon.  The results show that the error bound decreases with the number of iterations and the network width, which is expected. But It also shows that there is an additional term that depends on $\gamma^{T}$, which clearly indicates the difficulty of dealing with long-term memory problems where $T \rightarrow \infty$.
- In Theorem 6.3, the paper shows the error bounds for the Rec-NPG. The results show how the error bounds depend on a few things: 1) The number of iterations, where the error obviously decreases as we do more iterations. 2)The network size, increasing the network size, reduces the error. 3)The truncation horizon, we again see the $\gamma^{T}$. These results align with the analysis of the Rec-TD, but it’s valuable to see the consistency of the results when we move to the control setting.

### Weakness:
- In the abstract and again in the introduction and Section 5.1, it is mentioned that the paper studies multi-step TD; however, the theoretical analysis and the algorithm only use one-step TD errors.
- In Independent RNNs, the W matrix is diagonal. However, the analysis still involves matrix-vector multiplications with W, which unnecessarily complicates the maths.
- The notations used in Definition 3.1 were not properly introduced. It’s unclear what W(0) and U(0) mean.
- The theorems need more explanations on what they show.

---

> ### Author Response · Authors · 2025-07-23
>
> We would like to thank the reviewer for thoroughly reading our paper, and providing very valuable, positive, and constructive feedback, which helped us improve the overall quality of the paper. In the revised paper, we have highlighted the revisions in response to the comments of the reviewers in blue.
>
> Below we provide our responses to the specific points raised by the reviewer.
>
> - **Multi-step TD.** We thank the reviewer for the feedback. We have now removed "multi-step TD" in the revised version. Rec-TD uses a trajectory $\{(Y_t^k,A_t^k):t=0,1,\ldots,T-1\}$ of $T$ steps for each iteration of TD learning (Lines 8-14 in Algorithm 2), unlike conventional TD(0) that uses a single-step transition per TD-learning iteration. On the other hand, we agree with the reviewer that calling this multi-step TD learning can be misleading and confusing. As such, we have now removed the phrase "multi-step" in the revised version.
>
> - **Simplification of the notation.** We thank the reviewer for pointing this out. In Equation (8), we vectorize the hidden-to-hidden ($W_{ii}\in\mathbb{R}$ that corresponds to the $i$-th diagonal of **$W$**) and input-to-hidden weights for each neuron $i$, and provide the analysis based on this vectorized form throughout the paper. As such, we kindly note that the analysis involves the vectorized weights $\Theta_i=[W_{ii},U_i]$ and not matrices $(W,U)$. In order to simplify the notation even further, we have now revised Section 3 and clearly described the vectorization in Equation (8).
>
> - **Clarification of the random initialization in Definition 3.1.** We have now revised Definition 3.1 to clarify the random initialization, which is used to initialized IndRNNs used for both policy parameterization and policy evaluation.
>
> - **Additional discussions on the theoretical results.** We agree with the reviewer that including additional discussions on the theoretical results could improve the paper. To address this comment, we have now provided additional discussions after Theorem 5.4 and expanded Remark 6.4 to provide further intuition into the main theoretical results of the paper.
>
> - **Requested changes.**
>   - *Extension to fully-connected RNNs.* In order to address the comment by the reviewer, we have now added Remark 3.5 to discuss the challenges in dense recurrent layers. IndRNNs utilize a diagonal hidden-to-hidden layer $\mathbf{W}$, which was shown to be very effective in practice compared to conventional RNNs, GRU and LSTM in handling long-term dependencies in RL (Morad et al., 20203). In addition to its practical benefits, IndRNNs have theoretical niceties as well, as they enable (i) explicit characterization of the reference function class, and (ii) direct control and analysis of the spectral radius of $\mathbf{W}$. Both of these theoretical amenities are lost when $\mathbf{W}$ does not inherit a diagonal structure.
>   - *Element-wise computations.*  We have now revised the IndRNN using simplified update equations for each neuron $i$ (see Equation (6)) and their vectorized forms (see Equation (8)) to address the reviewer's comments.
>   - *Multi-step TD claims.* We have removed multi-step TD claims in the revision.
>   - *On the assumptions and limitations.*  We have now included a discussion on the assumptions (i) after Theorem 5.4 (on page 10) for Rec-TD, and (ii) in Remark 6.5 for Rec-NPG.
>   - *Mitigating the exploding gradient pathology.* We thank the reviewer for the reference. The effectiveness of the normalization step in the reference was shown for linear RNNs, i.e., $\varrho = \mathrm{Id}$. The extension to non-linear activation functions as in our setting is not immediate, therefore it is definitely a very interesting future research direction. Alternatively, preconditioning was shown to be very effective in (Martens and Sutskever, 2011) to mitigate this pathology by incorporating curvature information, which is another interesting direction to mitigate the curse of memory in RNNs. We have included a discussion on these future directions in the last section.

---

### Review · Reviewer_76rN · 2025-07-13

**Summary Of Contributions:**

This is a theory paper proposing a Natural Policy Gradient algorithm (actor-critic) for POMDPs and showing convergence guarantees under a number of assumptions.

The idea  and main contributions are the following:
* Parametrize the policy (actor) with an "Independently recurrent NN" (section 3), which simplifies the training dynamics and allows to get convergence rates,
* Use a TD-learning algorithm for the critic (section 5) and get convergence bound on the risk (th 5.4), ie on the approximation of the Q-function.
* Use NPG for the actor updates (section 6), which is now a non-stationary policy due to the POMDP, and prove error bounds on the best iterate (th 6.3)

**Audience:**

Yes

**Claims And Evidence:**

Yes

**Requested Changes:**

* improved writing and paper structure
* comments and intuitions for section 3
* experiments?

**Strengths And Weaknesses:**

## Strengths

* POMDPs are a hard problem and RNN-based algorithms are a standard approach that are lacking guarantees. This paper proposes a way to get such guarantees under some assumptions on the representation.

## Major comments

* This paper combines many technical ideas from previous work and relatively non-intersecting fields (POMDPs and MDP algorithms, neural network theory, optimization) so it is fairly difficult to understand well without the full background on all these topics. Typically, section 3.1 introduces "IndRNNs" based on Cayci & Eryilmaz (2024), as "a class of learnable mappings by RNNs in the so-called kernel regime, extending the function classes in previous literature on neural RL methods". That's quite a mouthful and also I am not sure I understand what this sentence says. What are "learnable mappings by RNNs"? This is the part of the background literature I know the less and I think it could deserve a little bit more intuition and explanations in plain English. This part is clearly instrumental in getting the convergence rates but it feels quite orthogonal to the algorithm itself so I think it could also be placed differently because at the moment it stands in between the problem formulation and the algorithm description in a perhaps awkward way.

* In the same vein, section 4 describes the overall algorithm and pushes the critic to section 5, but then section 6 comes back on the actor. Why not explaining the actor fully in section 4. I think the structure of the paper needs some more thoughts.

* Theorem 5.4: there is a middle term that does not go to 0 with K (only with m). Is that okay? How do you justify this? Also it is stated with "Rec-TD with max-norm regularization" but where is this regularization in Algorithm 2?

* You gave up on running any experiments? This is definitely a theory paper and I think it has enough contributions as such, but when one claims to bridge theory and practice, it is a lot more convincing if practice also aligns with the findings so the absence of experiments should be justified. Why did you not run any experiment?

## Minor

* I was confused in Algorithm 1 by "where Q=..." while there is no Q in the equation above. You want to say "where A=... and Q=..." because Q is in A.

---

> ### Author Response · Authors · 2025-07-23
>
> We thank the reviewer for thoroughly reading our paper, and providing very valuable, positive, and constructive feedback, which helped us improve the overall quality of the paper. In the revised paper, we have highlighted the revisions in response to the comments of the reviewers in blue.
>
> Below we provide our responses to the specific points raised by the reviewer.
>
> - **Characterization of the reference function class for POMDPs.** Thank you for your questions. In order to address them, in Section 3.1, we have considerably extended our discussion on the function class characterization of our IndRNN architecture. To that end, we first provide the necessary background for the basic MDP case with linear and (feedforward) neural function classes, and then, building on these ideas, we provide a more detailed description of the reference function class in the POMDP case.
>
>   **Placement of Section 3.1.** We completely understand your concern that it may distract the reader from the the algorithmic design. However, it is essential to have the definitions of the basic function classes, along with the introduction of our architecture, in order to provide the performance results in Sections 4 and 5. In order to alleviate this concern, we have added a comment right before Section 3.1 to allow readers to skip the details if they would like to focus on the algorithms. We hope this resolves your concern.
>
> - **Structure of the paper.** We would like to thank the reviewer for the feedback. Section 4 is aimed to provide a high-level (encapsulating) description of the overall actor-critic method, with a clear roadmap towards the discussion on policy evaluation (Section 5) and policy optimization (Section 6). Here, we aim to motivate the development of Rec-TD and Rec-NPG algorithms at a high level within the actor-critic framework, and guide the readers to the discussions and results in each part. In order to address the reviewer's feedback, in the revision, we have now further clarified these points, and provided further directional remarks to guide the reader.
>
> - **Regarding Theorem 5.4.** The term $O(1/\sqrt{m})$ in Theorem 5.4 is related to the expressiveness of the IndRNN, where we denote the neural network width as $m$. A larger $m$ implies a more expressive neural network with a smaller approximation error for approximating the reference function class $\mathscr{F}$ in Definition 3.3 using finite-width networks (see the last term on the RHS of Equation (11) and also Lemma B.2). As such, the corresponding term in Theorem 5.4 vanishes at a rate $\mathcal{O}(1/\sqrt{m})$. Similar error term that stems from using a finite-width neural network appears in neural-TD literature (Cai et al., 2019; Wang et al., 2019) for the case of MDPs.
>
>   **Max-norm projection.** In Algorithm 5.2, Line 15, $Proj_{\Omega,m}$ applies max-norm projection to the parameters to ensure that $\Theta(k)\in\Omega_{\rho,m}$ throughout the training procedure $k\geq 1$. In the revision, we have now included all the details of max-norm projection in Section 3.2, and discussed the use and importance of the max-norm projection in Section 3.2 and (extended) Remark 5.5 to provide intuitive and practical insights.
>
> - **Numerical results.** As you correctly noted, our core contributions are on the theoretical side of solving RL for POMDPs by using RNN architectures. In Appendix C, we have provided some empirical results to corroborate the theoretical results on the impact of various system parameters on the performance of the core policy evaluation algorithm Rec-TD. More extensive experimental studies is beyond the scope of our current focus, but is definitely a good and important direction of future research that we hope to pursue.

---

### Decision · Action_Editor_ZBVv · 2025-09-25

**Recommendation:** Accept as is

**Additional Comments:**

The reviewers all agree that their concerns were addressed, and the paper improved over the review process.

**Audience:**

Yes

**Audience Explanation:**

RNNs are broadly of interest to TMLR, and the studied properties of RNNs provide some practical insights.

**Claims And Evidence:**

Yes

**Claims Explanation:**

The paper has theoretical statements that are well-supported. The clarity of the document improved during the review process.